# Understanding resonant charge transport through weakly coupled single-molecule junctions

James O. Thomas [1,2,6]*, Bart Limburg[1,2,6]*, Jakub K. Sowa[2,6], Kyle Willick[3], Jonathan Baugh[3], G. Andrew D. Briggs [2], Erik M. Gauger[4], Harry L. Anderson [1]* & Jan A. Mol[2,5]*

Off-resonant charge transport through molecular junctions has been extensively studied since the advent of single-molecule electronics and is now well understood within the framework of the non-interacting Landauer approach. Conversely, gaining a qualitative and quantitative understanding of the resonant transport regime has proven more elusive. Here, we study resonant charge transport through graphene-based zinc-porphyrin junctions. We experimentally demonstrate an inadequacy of non-interacting Landauer theory as well as the conventional single-mode Franck–Condon model. Instead, we model overall charge transport as a sequence of non-adiabatic electron transfers, with rates depending on both outer and inner-sphere vibrational interactions. We show that the transport properties of our molecular junctions are determined by a combination of electron–electron and electron-vibrational coupling, and are sensitive to interactions with the wider local environment. Furthermore, we assess the importance of nuclear tunnelling and examine the suitability of semi-classical Marcus theory as a description of charge transport in molecular devices.

[1] Department of Chemistry, University of Oxford, Chemistry Research Laboratory, Oxford OX1 3TA, UK. [2] Department of Materials, University of Oxford, Parks Road, Oxford OX1 3PH, UK. [3] Institute for Quantum Computing, University of Waterloo, Waterloo, ON N2L 3G1, Canada. [4] SUPA, Institute of Photonics and Quantum Sciences, Heriot-Watt University, Edinburgh EH14 4AS, UK. [5] Department of Physics, Queen Mary University, London E1 4NS, UK. [6]These authors contributed equally: James O. Thomas, Bart Limburg, Jakub K. Sowa. *email: james.thomas@chem.ox.ac.uk; bart.limburg@chem.ox.ac.uk; harry.anderson@chem.ox.ac.uk; j.mol@qmul.ac.uk

A quantitative understanding of the mechanism of charge transport in molecular junctions is not only vital for the future development of functional molecular electronic circuits[1] but can also shed light on the electron transfer reactions in areas such as photochemistry, electrochemistry and catalysis. The off-resonant transport regime, in which the molecular energy levels are far from the Fermi level of the electrodes, is well described by non-interacting scattering approaches[2]. These approaches are epitomised by Landauer theory, in which the molecule is reduced to a scattering centre with an energy-dependent transmission spectrum. However, in weakly coupled molecular junctions, when one of the molecular energy levels falls within the bias window between the Fermi levels of the electrodes, the overall charge transport takes place through a different mechanism. An electron tunnelling from the source electrode localises on the molecule for a short time before tunnelling into the drain. In contrast to redox molecular junctions[3] (in which the charging/discharging of the molecule has no direct contribution to the current), the current that flows through the molecular junction during resonant transport in a weakly coupled junction is a result of these sequential electron transfers to (i.e., a reduction process) and from (i.e. an oxidation process) the molecule. As both the electron occupancy and the equilibrium geometry of the molecule and its local environment change upon an electron transfer event, the electron–electron and electron-vibration interactions can no longer be ignored[4]. In order to model the resonant transport through the junctions we shall employ a rate-equation approach (see Supplementary Note 2) which captures the effects of the aforementioned interactions.

The expression for the net current through a weakly coupled molecular junction has a well-known form[4–8]:

$$I = |e| \frac{\gamma_{ox}^S \gamma_{red}^D - \gamma_{red}^S \gamma_{ox}^D}{\gamma_{red}^S + \gamma_{red}^D + \gamma_{ox}^S + \gamma_{ox}^D},\qquad(1)$$

where $e$ is the elementary charge and $\gamma_{red/ox}^l$ denote the rates of (non-adiabatic) electron transfers at each electrode (l = S/D for the source and drain electrode, respectively). The rates in equation 1 are given by:

$$\gamma_{red}^l = (2 - \Omega) \frac{\Gamma_l}{\hbar} \int f_l(\epsilon) k_{red}(\epsilon) d\epsilon,\qquad(2)$$

$$\gamma_{ox}^l = (1 + \Omega) \frac{\Gamma_l}{\hbar} \int (1 - f_l(\epsilon)) k_{ox}(\epsilon) d\epsilon,\qquad(3)$$

where $\Gamma_l$ is the electronic coupling to the source/drain electrode and $f_l(\epsilon)$ is the Fermi–Dirac distribution in the electrode l. The presence of the factors $\Omega$ in equations 2 and 3 is a direct consequence of strong electron–electron interactions which, at a given gate voltage, preclude changing the charge state by more than one. Therefore, if tunnelling occurs into an unoccupied orbital (LUMO) (e.g. the N/N+1 transition, where N is the total number of electrons on the molecule in the neutral state) two possible pathways exist for reduction – an electron of either spin can tunnel from the electrode onto the molecule. Only one possible path exists for the subsequent oxidation as the unpaired electron (in what is now the SOMO) tunnels out of the molecule and into the electrode. Conversely, if tunnelling occurs into a singly occupied orbital (e.g. the N−1/N transition) the opposite is the case: only electrons of the opposite spin to that on the molecule can reduce the molecule, but electrons of either spin can subsequently tunnel out from the molecule into the leads. When only a single spin-degenerate level is involved in transport then the number of possible transitions is accounted for by setting $\Omega$ to 0 for the N/N+1 transition or 1 for the N−1/N transition, as discussed in Supplementary Note 2 and in detail elsewhere[9].

Finally, $k_{red/ox}$ denote the molecular densities of states (DOS) associated with the corresponding electron transfers. They are primarily determined by the structural reorganisation of the molecule and its environment upon electron transfer and therefore account for the effects of electron-vibrational coupling[10]. As we shall discuss, the molecular DOS should generally also account for the lifetime broadening of the electronic states (which may be attributed to the time-energy uncertainty relationship) which is especially important at low temperatures.

At higher temperatures (and when the electronic degrees of freedom interact predominantly with the outer-sphere environment), the molecular DOS can be obtained using the semi-classical Marcus theory (MT) which treats the nuclear degrees of freedom classically and disregards lifetime broadening (see below). While typically applicable at ambient conditions (as confirmed experimentally[11,12]) this approach is expected to break down at cryogenic temperatures where lifetime broadening and the quantum nature of the vibrational motion become relevant. Previous studies have estimated the molecular DOS at low temperatures typically by assuming coupling of the electronic degrees of freedom to a single vibrational mode with limited success (often also disregarding the effects of electron–electron interactions or those of lifetime broadening)[13–15].

In this work, we go beyond such single- (or many-)mode Franck–Condon models[16,17], by accounting for the outer-sphere vibrational coupling (to the substrate on which the molecule is deposited). These interactions are usually ignored in the molecular-junction setting despite the fact that the recent experimental studies have highlighted a significant contribution from the dielectric substrate to the reorganisation energy of single molecules[18]. We investigate the importance of nuclear tunnelling and experimentally assess the applicability of MT (and its refinements) in the considered systems. We also demonstrate that the theoretical approach used here, combined with DFT calculations, can be used to elucidate the mechanism of the experimentally observed resonant charge transport in the studied weakly coupled single-molecule junctions.

## Results

**Molecular devices**. The device architecture we use as a platform to study electron transfer is shown schematically in Fig. 1a and is described in detail in the Methods, and shown in Supplementary Fig. 1. Briefly, we fabricate graphene nanogaps that comprise pairs of source and drain electrodes spaced by 1–2 nm using feedback-controlled electroburning[19–22]. Zinc porphyrin molecules, functionalised with anchor groups that have been designed to bind to the graphene electrodes via π–π stacking and van der Waals interactions[23] (Fig. 1b), are deposited from solution. 3,5-Bis(trihexysilyl)phenyl aryl groups increase the solubility of the porphyrin and prevent aggregation, however, we do not expect them to directly contribute to the charge transport, as DFT calculations indicate that during the oxidation/reduction of the molecular species the additional charge density is localised on the porphyrin ring and the pyrene anchor groups (although the aryl groups may affect the molecule-lead and outer sphere electronic-vibrational coupling). A gate electrode is used to adjust the energy of the molecular levels relative to the Fermi levels of the source and drain electrodes. The presence of the gate electrode is crucial as it allows us to investigate the resonant transport regime.

**Low-temperature measurements**. Figure 1c shows a low-temperature (3.5 K) conductance map for such a device (device **A**) measured as a function of bias and gate voltage (see Supplementary Fig. 15 for the entire stability diagram). Within most of the map the current is Coulomb-blocked, indicating that π-π

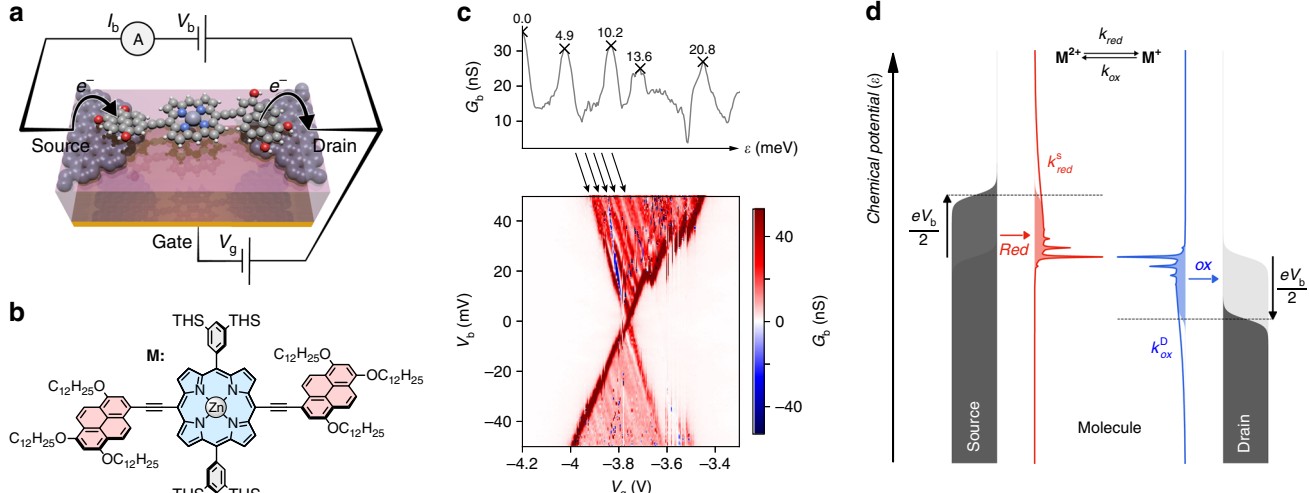

**Fig. 1** Charge-transport characteristics of a graphene-porphyrin single-molecule junction. **a** Schematic representation of our device architecture: nanometre-separated graphene source and drain electrodes are used to contact the molecule, and a local gate electrode separated from the molecule by a thin layer of HfO$_2$ (10 nm thick) is used to shift the molecular energy levels. For clarity, the bulky side-groups are not shown. **b** The molecule M used in this study comprises of a porphyrin core (blue), with solubilising aryl side-groups on two of the porphyrin *meso*-positions (grey), and π-stacking anchor groups on the other two *meso*-positions (red); here THS is trihexylsilyl. **c** Charge stability diagram showing the differential conductance ($G_b$) as a function of bias voltage ($V_b$) and gate voltage ($V_g$) at 3.5 K; the actual gate voltage experienced by the molecule is only a fraction of the applied gate voltage because of the drop across the HfO$_2$ layer. The top panel shows the differential conductance of the top triangle as an average along the lines indicated by the arrows running parallel to the edge of the triangle. **d** Schematic representation of current flowing through our single-molecule transistor. The molecular DOS for reduction and oxidation processes are shown in red and blue, respectively, with electron-transfer rates shown as coloured areas. The Fermi-Dirac distributions $f_S$ and $f_D$, are shown as the grey areas for source and drain, respectively. At negative bias voltage, electrons tunnel sequentially from the source via the molecule into the drain. For convenience, the bias voltage is drawn as applied symmetrically across the source and drain electrodes

stacking leads to weak molecule-electrode coupling. In addition, we observe a high conductance region in which sequential electron transfers take place. As the transition considered here is the second closest in terms of the gate potential to the Fermi level of the graphene leads (see the full charge stability in the Supplementary Note 5), it is likely to be the transition between the *N–2* and *N–1* charge states (where *N* charge state corresponds to the neutral molecular species, *i.e.* the M$^{2+}$/M$^+$ transition, see Supplementary Fig. 3 for the relevant frontier orbitals). Our assignments of the charge states are confirmed by observing the high-current corner of each transition.[9] Inside the sequential tunnelling region, we observe lines of increased conductance (Fig. 1c), which are spaced equally apart. We are able to assign these conductance lines to vibrational excitations of the molecule during the charging process, in line with previous studies[13,14,24]. At low bias resonant transfer occurs between the vibrational ground states of both charge states. As the bias voltage is increased, electron transfer onto the molecule can be accompanied by a vibrational excitation.

The assignment of the conductance lines to molecular vibrations, as opposed to *e.g.* density of states (DOS) fluctuations in the graphene[25], is robust despite the presence of some imperfections in the experimental data (such as jumps in the edges of the Coulomb diamond) for several reasons. Firstly, the line graph in Fig. 1c shows the data in the high conductance region averaged along a series of lines that run parallel to the edge of the Coulomb diamond and plotted as a function of potential: the peaks we observe would not be present if the lines did not run, at least approximately, parallel to the edges of the high conductance region. Furthermore, fluctuations in the DOS do not typically give stepwise increases in the current, but rather regions of increased conductance alternated by regions of negative differential conductance, which we do not observe. The spacing between the lines is approximately equal, which is a

feature of molecular modes and overtones, and unlikely to be present in DOS fluctuations. Finally, we found the same equally spaced conductance lines in another device (device **E**, see Supplementary Note 6). From Fig. 1c we calculate the average spacing between the lines measured for device **A** to be 4.9 ± 0.3 meV, which is in a rough agreement with DFT calculations which predict the presence of a strongly-coupled low-energy vibrational mode (at 6.0 meV, see Supplementary Fig. 6). We note however that any assignment should be treated with caution due to strong anharmonic effects often observed for low-frequency molecular modes.

The current–voltage trace of device A measured on resonance (Fig. 2a) reveals an asymmetry between the current at positive and negative bias voltages. The potential drop across the molecule is almost symmetric: $\alpha_S = C_S/C_{tot} = 0.45$, where $C_s$ is the capacitance to the source and $C_{tot}$ is the sum of the capacitances to the source, drain and gate. Therefore the current rectification is not due to an asymmetric potential drop across the molecule[26]. Instead it is a direct result of electron–electron interactions in the presence of asymmetric molecule-electrode couplings and spin degeneracy[7] (accounted for by $\Omega$), and can be inferred from equation 1–3. The current rectification ratio will be between 1 (for symmetric coupling, $\Gamma_S \approx \Gamma_D$) and 2 (for strongly asymmetric coupling, $\Gamma_S \gg \Gamma_D$ or vice versa), and will alternate along with $\Omega$ for adjacent charge states. The current rectification observed in our experiments cannot be explained if the electron–electron interactions are ignored (as within the non-interacting Landauer approach) or in the case of strong coupling between the molecule and the electrodes (where the energy uncertainty associated with the lifetime of the electronic states is greater than the energy required to change the charge state of the molecule).

We proceed to quantitatively describe the observed charge transport by accounting for both lifetime broadening and the

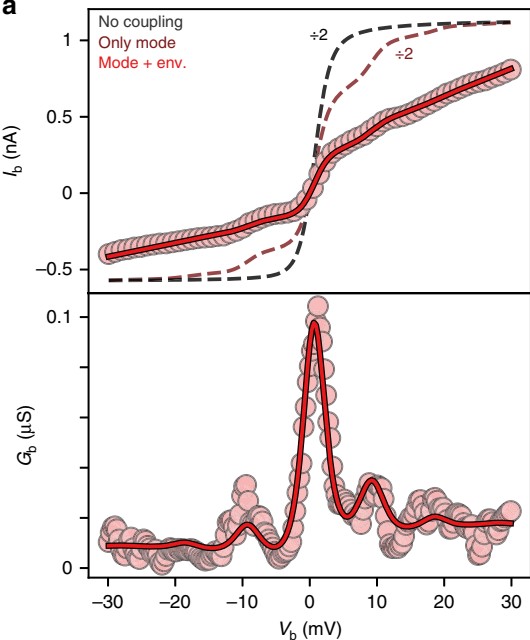

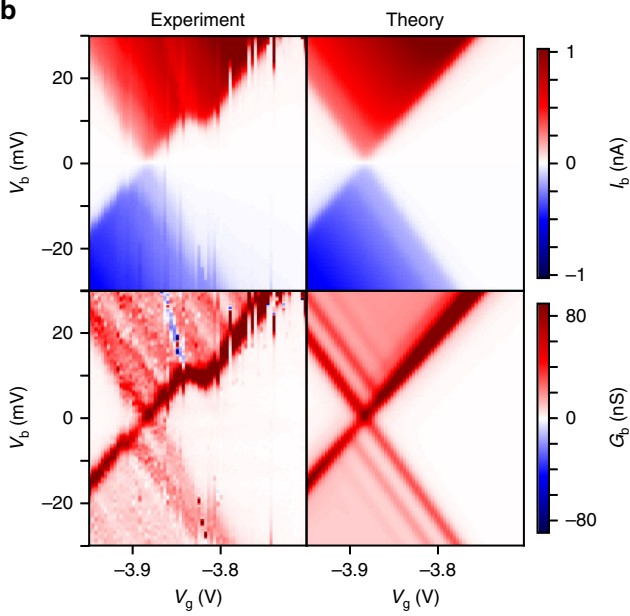

**Fig. 2** The contributions of inner and outer sphere vibrational interactions in device A. **a** Current ($I_b$) and differential conductance ($G_b$) as a function of bias voltage ($V_b$) of device A (circles) at 5 K, corresponding fit to our model (red line), and corresponding curves without environmental coupling (dark red line) or without vibrational coupling (black line). **b** Experimental current (left top) and differential conductance (left bottom) maps as a function of bias and gate voltage of device A, and reconstructed maps (right) from the parameters used to fit the IV trace in a). At higher (positive) bias we observe switching in the stability diagram most likely resulting from a nearby charge trap[30]. This effect is however inconsequential to the phenomena discussed here

influence of the vibrational environment in our quantum-mechanical expression for the molecular DOS[5]:

$$k_{red/ox}(\epsilon) = \frac{1}{\pi} \mathrm{Re} \int_0^\infty e^{\sigma i(\epsilon - \mu)t/\hbar} e^{-t/\tau} B(t)dt, \qquad (4)$$

where $\tau = 2\hbar(\Gamma_S + \Gamma_D)^{-1}$ is the lifetime of the electronic state,

and $\mu$ the energy level of the molecule. The sign $\sigma$ is either +1 for reduction or –1 for oxidation. The phononic correlation function, $B(t)$, which can be thought of as a time-dependent Frank–Condon factor that describes the nuclear dynamics accompanying electron transfer[27], is given by:

$$B(t) = \exp\left[\int \frac{J(\omega)}{\omega^2}\left(\coth\left(\frac{\omega}{2k_B T}\right) \times (\cos\omega t - 1) - i\sin\omega t\right)d\omega\right], \qquad (5)$$

where $J(\omega) = \sum_q |g_q|^2 \delta(\omega - \omega_q)$ is the spectral density for vibrations with frequencies $\omega_q$ and electron-vibration coupling strengths $g_q$; $k_B T$ is the thermal energy.

Now having introduced our model, we begin by fitting the differential conductance of device **A** on resonance to equation **1**, Fig. 2a (bottom panel), with $\Gamma_S$, $\Gamma_D$, $\omega_q$ and $g_q$ as the fitting parameters. We found that the low-bias electron transfer is dominated only by a single molecular vibrational mode with energy $\hbar\omega_q = 4.2$ meV and Huang–Rhys parameter $S_q = g_q^2/\hbar^2\omega_q^2 = 0.4$. However, a spectral density consisting of only this single mode (the usual Franck-Condon model)[16] cannot reproduce the experimental data. Only if we account for the coupling to the substrate, do we find a good agreement with the empirical data, as shown in Fig. 2a (top panel). We model this outer-sphere background using a structureless super-Ohmic spectral density with an exponential cut-off. Such a spectral density can be used to describe the (deformation) coupling of a localised charge to bulk phonons[28–30] and constitutes the simplest description of this environmental contribution, see Supplementary Note 4.

The complete fit therefore comprises two additional parameters to describe the environment: the corresponding reorganisation energy, $\lambda_o$, and the cut-off phonon frequency, $\omega_c$. From the fit we obtain $\lambda_o = 26$ meV and $\hbar\omega_c = 8.3$ meV (we note that only the low-frequency part of the outer-sphere background can be extracted from the low-bias measurements considered here). The overall spectral density therefore contains both an inner sphere contribution, corresponding to structural reorganisation of the molecule, and an outer sphere contribution from the surrounding dielectric environment. Molecule-electrode coupling leads to a lifetime broadening of the conductance peaks: $\hbar/\tau = 0.31$ meV. Omitting lifetime broadening leads to a ~30% over-estimation of the zero-bias conductance at 5 K (see Supplementary Note 6). The validity of our approach is further substantiated by the fact that using the parameters obtained from fitting a single differential conductance trace on resonance, we can calculate the entire current map as function of bias and gate voltage which shows very good agreement with the experimental data, as shown in Fig. 2b.

**Temperature-dependence.** We proceed to consider the temperature dependence of the observed transport behaviour. As shown in Fig. 3a, we can successfully fit the resonant current–voltage traces between 5 K and 70 K with the spectral density extracted above, i.e. using the extracted parameters: $\omega_q$, $S_q$, $\lambda_o$ and $\omega_c$. It is necessary, however, to re-fit $\Gamma_{S/D}$ at each temperature to account for apparent changes in the exact nature of the molecule-electrode contact as the junction is warmed up. Experimentally, we find that as the temperature increases the resonant conductance decreases and the structure of the differential conductance is washed away. This can be explained by the simultaneous thermal broadening of the Fermi distributions in the leads and the molecular DOS, $k_{red/ox}$.

At higher temperatures, $k_B T \gg \hbar\omega, \hbar/\tau$, it is possible to simplify equation **4** by disregarding lifetime broadening and considering a high-temperature limit within the phononic

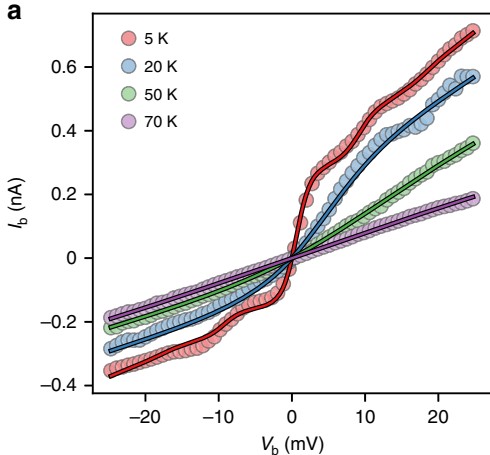

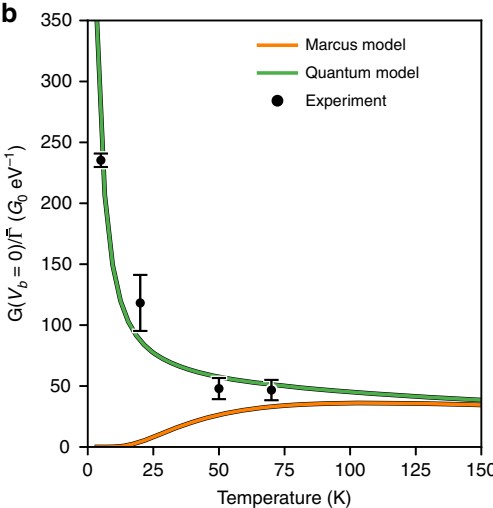

**Fig. 3** Temperature-dependent charge-transport in device **A**. **a** *IV* traces of device **A** at various temperatures (circles) and the global fit (solid lines) where all parameters except the molecule-electrode couplings are shared. **b** Zero-bias conductance normalised by ($\bar{\Gamma} = \Gamma_S \Gamma_D / (\Gamma_S + \Gamma_D)$) as a function of temperature. Black circles: experimental values with error bars (95% confidence interval) obtained by propagating a random 0.2 pA error on current measurements with the uncertainty on $\Gamma_S$ and $\Gamma_D$ obtained from the fits to the quantum model shown in **a**. Orange line: Marcus model using $\lambda = 27.6$ meV. Green line: Quantum model using one mode and a broad background ($\hbar\omega = 4.1$ meV, $S = 0.4$, $\lambda_o = 26$ meV, $\omega_c = 8.3$ meV)

correlation function[24]. This yields the previously discussed MT in which the molecular DOS takes the familiar classical form[11,31,32]:

$$k_{red/ox}(\epsilon) = \sqrt{\frac{1}{4\pi\lambda k_B T}} \exp\left[-\frac{(\lambda \pm (\epsilon - \mu))^2}{4\lambda k_B T}\right], \quad (6)$$

where $\lambda = \lambda_i + \lambda_o$ is the total reorganisation energy. From the fit to device **A** in Fig. 2a we can calculate the total reorganisation energy as $\lambda = \hbar S_q \omega_q + \lambda_o = 27.6$ meV.

To assess the applicability of MT to device **A**, we consider the zero-bias conductance scaled by the molecule-electrode coupling (to correct for the variations in $\Gamma_{S/D}$). In Fig. 3b we plot the (scaled) zero-bias conductance observed experimentally for device **A** as well as calculated using the quantum and Marcus models (using parameters extracted from the fit in Fig. 2). At low temperature, the quantum and Marcus approaches display opposite trends of zero-bias conductance *vs.* temperature.

Thermal broadening of the Fermi distributions of electron energies in the leads, and increased population of excited vibrational states, lead the quantum model to display a zero-bias conductance that decreases with increasing temperature, in agreement with what is observed experimentally. In MT, electron transfer is driven by thermal fluctuations and consequently the zero-bias conductance within the considered range increases with temperature. At low temperature the MT electron transfer rates, and therefore conductance, vanish since this approach does not account for nuclear tunnelling (*i.e.* overlap between the vibrational wavefunctions in the classically forbidden region is neglected)[32]. Comparison of the experimental data with the quantum and MT models demonstrates the importance of incorporating this effect. A quantum mechanical description of electron transfer is clearly necessary at low temperature, especially below 50 K, and continues to be an accurate description of electron transfer across the whole temperature range. As expected from a theory developed as a high-temperature limit, MT constitutes an increasingly accurate description of the data as temperature increases, and by inspecting Fig. 3b we can infer that there will be a temperature at which the quality of a fits to a quantum or MT-based will converge. This temperature will be dependent on the reorganisation energy, $\lambda$ (*i.e.* larger $\lambda$ leads to a higher convergence temperature), and the exact details of the quantum spectral density. We expect that, in general, MT is an adequate model for electron transfer in weakly coupled molecular systems at 298 K.

**High-bias studies**. To further explore the correspondence of quantum and semi-classical descriptions of electron transfer, we compare the performance of MT with our quantum model for three devices, **B**–**D** (fabricated with a 300 nm $SiO_2$ gate dielectric, see Methods) at 77 K. Since devices **B**–**D** were measured within a larger bias voltage range it is now necessary to incorporate all the molecular vibrational modes in the quantum analysis of the electron transport. Therefore, in what follows, the overall spectral density in equation 5 comprises all molecular vibrational modes as well as a broad background, $J_{bg}(\omega)$, which phenomenologically accounts for the dielectric substrate:

$$J(\omega) = \sum_q \left|g_q\right|^2 \delta\left(\omega - \omega_q\right) + J_{bg}(\omega). \quad (7)$$

The frequencies and coupling strengths of the molecular modes were calculated using DFT and correspond to an inner-sphere reorganisation energy of $\lambda_i = \hbar \sum_q \omega_q S_q$ of 67 meV for the *N–1/N* transition (see Supplementary Note 3 for details of the calculation). The *N–1/N* transition is considered the most likely assignment as the closest transition to the Fermi level of the electrodes, and confirmed by observation of the highest current corner[9]. The background is again modelled as a structureless super-Ohmic spectral density. These outer and inner-sphere contributions are plotted in Fig. 4a. Figure 4b shows the comparison between the quantum and MT molecular DOS calculated for the instructive values of $\lambda$. It demonstrates that at 77 K, the temperature at which devices **B**–**D** were measured, the quantum DOS extends over a broader energy range than their Marcus counterparts. As molecular vibrations from low-energy bending motion (a few meV) to $sp^2$ C–C bond stretches and C–H bond stretches (~200 meV and 400 meV respectively) are taken into account, we have maxima in the quantum DOS either side of the peak in the Marcus rates, and the symmetry around $\epsilon = \mu \pm \lambda$ is not present.

We find that the experimental charge transport data for devices **B**–**D** at 77 K can be described by MT since at this temperature the errors in the fits of the *IV* characteristics are not particularly large

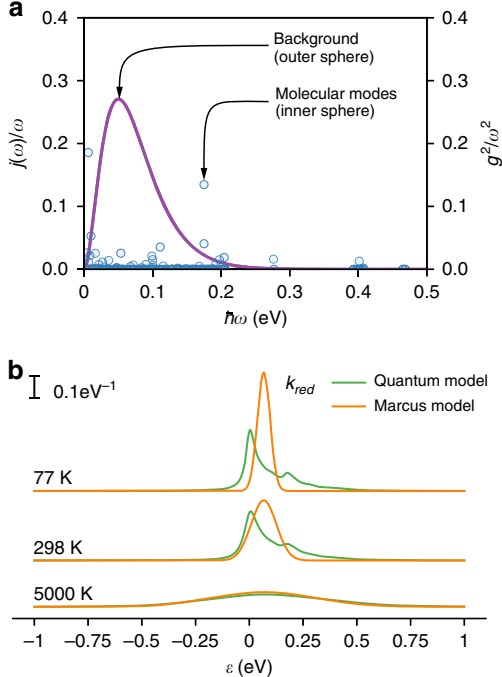

**Fig. 4** Comparison of rate constants for quantum and Marcus models. **a** Spectral density used for the quantum model consisting of a background and individual molecular modes calculated by DFT. Background parameters: $\lambda_o = 25$ meV, $\hbar\omega_o = 25$ meV. **b** Calculated $k_{red}$ at the drain electrode as a function of chemical potential for 3 different temperatures for the quantum and Marcus models. Parameters: quantum: $\lambda_o = 25$ meV, $\hbar\omega_c = 25$ meV, $\lambda_i = 67$ meV; Marcus: $\lambda = 92$ meV

compared to the magnitude of the current (as shown in Fig. 5a). However, there are features in the data that are not captured by this approach that we must explain if we wish to develop a detailed and physical understanding of the mechanism of charge transport that is valid over a wide temperature range and robust to changes in the molecular structure. In particular, conductance at low-bias voltages is underestimated since MT treats the nuclear dynamics classically. At low bias voltages, the barrier for electron tunnelling cannot be overcome solely due to thermal fluctuations of the environment, resulting in very low electron-transfer rates. In reality, however, electron transfers at low bias are dominated by nuclear quantum mechanical tunnelling and consequently electron transfer can occur relatively efficiently. This shortcoming of MT can be partially mitigated by expanding the phonon correlation function to higher order[5,33], or by coarse-graining low- ($\hbar\omega_q \ll k_B T$) and high-energy ($\hbar\omega_q \gg k_B T$) vibrational modes as is done in the Marcus-Levich-Jortner theory[33,34]. As shown in Supplementary Note 8, for some devices such approaches rectify the limitations of MT, (at the expense of additional fitting parameters) and highlight the non-classical mechanism of electron transfer in these relatively high-temperature (for single-molecule devices) conditions. Our experimental data, however, are generally better described by our fully quantum mechanical treatment involving both inner and outer sphere reorganisation, as discussed above.

The current–voltage traces of devices B–D in Fig. 5a are therefore fitted using the quantum approach and with spectral density given in equation 7, taking only $\Gamma_S$, $\Gamma_D$ and $\lambda_o$ as free fitting parameters. The cut-off phonon energy, $\hbar\omega_c$, is fixed at 25 meV, and we expect this parameter to be intrinsic to the SiO$_2$ substrate (see Supplementary Note 4 and 7 for the dependence of the fitting on $\hbar\omega_c$). We obtain outer-sphere reorganisation

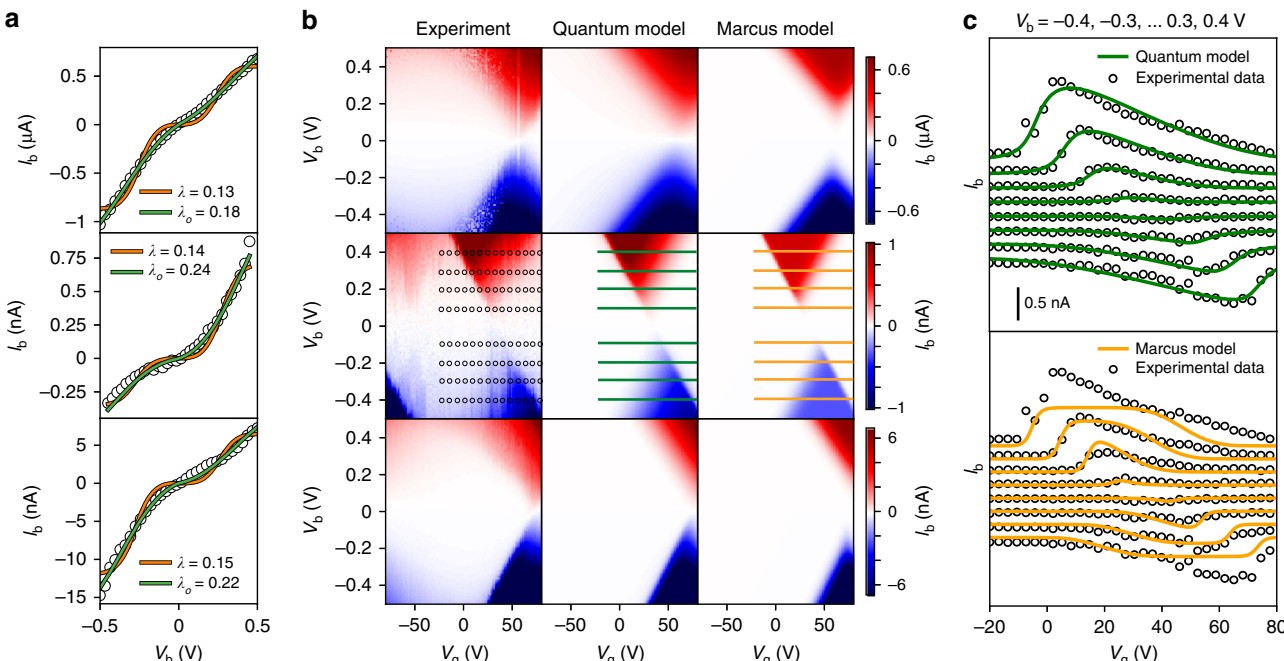

**Fig. 5** Investigation of devices B–D at 77 K. **a** IV traces at resonance of 3 devices (B, C, D) of molecule M on SiO$_2$ dielectric at 77 K (data from ref. [22]) and the fit to the Marcus model (orange) and the quantum model (green). For clarity, fewer experimental data points were shown. **b** Charge stability diagrams of the same 3 devices as in c, and the reconstructed stability diagrams from the fits in c according the quantum model, or the Marcus model. **c** Gate traces ($I_b$ vs. $V_g$) at multiple values of $V_b$ (indicated by horizontal lines in b)) for device C to compare experimental data and fits to the quantum model (upper panel, green) and Marcus model (lower panel, orange). $I_b$ values for consecutive gate traces are offset by 0.2 nA for clarity. Gate trace comparisons for devices B and D are displayed in Supplementary Note 7

energies of $\lambda_o = 180$, 240, and 220 meV for devices **B**–**D**, respectively. We assign the relatively large variation in outer-sphere reorganisation energy to small variations in the distance of the porphyrin from the dielectric surface. In Supplementary Note 4, by modelling the porphyrin as a rectangle with uniformly distributed charge, we estimate that the above values of $\lambda_o$ correspond to the porphyrin molecules being roughly up to 0.71, 0.51 and 0.58 nm away from the SiO$_2$ dielectric substrate[18,35], matching half the height of a monolayer of these porphyrin molecules on an HOPG surface[23]. Our approach successfully accounts for the asymmetries of the observed transport characteristics (both with respect to the applied bias and gate voltage), this is shown in Fig. 5c. We further note that, in the case of device **C**, the low-bias current is very strongly suppressed. This is an example of a Franck-Condon blockade[16], and is very well captured by our theoretical model.

It is important to note that in fitting the data we obtain a reorganisation energy from either a MT or quantum model fit that represents a lower bound of the true value. The experimental bias window is limited to a few tens of millivolts at 5 K to a few hundred millivolts at 77 K due to the instabilities of nanoscopic junctions and therefore we do not probe the full energy spectrum of electron–phonon coupling. Vibrational modes that lie above the probed spectrum could contribute to the overall reorganisation energy but their contribution to the measured current is not captured in the experimental data. In the fitting of experimental data to MT a balance exists: the current suppression at low bias can be alleviated by decreasing $\lambda$, however this leads to an early current plateau at high bias. Conversely, an increase in $\lambda$ removes the high-bias current plateau, but greatly exacerbates the current suppression at low bias. Therefore, fitting will commonly result in the centre of the Gaussian-shaped rate constants being placed roughly half-way on the experimental applied bias voltage in order to minimise both of these unphysical effects. In the absence of a saturation of the current in experimental data (which we never observed in any of our data), MT will therefore always give an underestimation of the reorganisation energy. The quantum model includes nuclear tunnelling, and the artificial Frank-Condon blockade is not present, and we therefore do not have to compromise between underestimating the low-bias current and an early plateau. Consequently, the quantum model generally gives higher reorganisation energies that more closely match the true value. However, in the absence of saturation in the current, due to the absence of data outside the bias window, the quantum model also results in a low bound for the reorganisation energy.

In addition to devices **B**–**D**, in Supplementary Note 7 we present data for 9 more devices measured at 77 K, comprising porphyrin molecules that differ only by the $\pi$-anchoring group (chemical structures are given in Supplementary Note 1). We similarly calculated $\lambda_i$ for the molecular species using DFT methods (see Supplementary Note 3), and then fit our data to three parameters: $\Gamma_S$, $\Gamma_D$ and $\lambda_o$. The results (presented in Supplementary Note 7) show the transport behaviour of these devices can also be successfully explained using our quantum model. The values of $\lambda_o$ obtained for these devices are in the range of 110–250 meV. As expected, the effectiveness of the theoretical approach used here does not depend on the exact chemical structure of the molecular species. These results further emphasise the importance of the broader molecular environment, since even in our simple device-architecture, small changes in the molecule-substrate distance lead to large changes in reorganisation energy, see Supplementary Fig. 22. This in turn results in significant variations in the electron-transfer rates and current–voltage characteristics. In order to address the issues of reproducibility in molecular-scale electronics, the results show us that we must look for ways to control the local environment surrounding the molecule of interest.

## Discussion

In this work, we have studied resonant charge transport through zinc-porphyrin molecular junctions. We have demonstrated that the non-trivial conductance properties of these systems can be explained by a combination of the outer- and inner-sphere vibrational coupling and the electron–electron interactions. In contrast, neither the conventional Landauer theory nor the single-mode Franck–Condon model provide an accurate theoretical description of the experimentally observed charge transport. We have further shown that at cryogenic temperatures (below 77 K), Marcus theory constitutes a less accurate description of the charge transport mechanism due to the importance of nuclear tunnelling under these conditions. Conversely, our quantum transport model which, besides the electron-vibrational interactions, also accounts for lifetime broadening and spin-degeneracy of the electronic levels, yields good agreement with the experimental data at all temperatures. An examination of the temperature dependence of the quantum and Marcus theories suggests that correspondence between the two approaches should be reached in our devices at some point above 100 K, but will depend on the overall value of the reorganisation energy. We have shown that all the ingredients of our quantum model are necessary to develop a quantitative description of resonant transport through weakly coupled single-molecule junctions, especially at low temperature. Therefore, we believe that the theoretical description validated here should be broadly applicable throughout the field of molecular electronics.

Our results further demonstrate that in the design of functional molecular technologies such as molecular transistors, diodes and thermoelectric materials, attempts must be made to precisely control the (often ignored) molecular outer-sphere environment. This could be achieved by, for example, synthesising supramolecular assemblies that isolate the molecular structure from the local environment[36,37]. Finally, we have also shown that single-molecule junctions can act as a tool to unravel the mechanism of individual electron transfers in molecular systems. This opens the door towards precise single-molecule experimental investigations of the influence of various liquid, solid or supramolecular environments on the rates of heterogeneous electron transfers. A comprehensive understanding of the influence of the local environment on electron transport could have significant impact on improving reproducibility in single-molecule electronics or optimising the performance of thin-film organic electronic devices.

## Methods
**Device architecture**. Device **A** and device **E** (Supplementary Note 6) are fabricated using the procedure outlined in the following sections. The details of the device fabrication as well as the experimental transport data for devices **B**–**D** and **F**–**M** have been published previously[23].

**Substrate fabrication**. Devices **A** and **E** were fabricated on a degenerately $n$-doped silicon wafer with a 300 nm thick layer of thermally-grown SiO$_2$. A 3 μm wide local gate was defined by optical lithography and electron-beam evaporation of titanium (10 nm) and gold (30 nm). A 10 nm layer of HfO$_2$ was then deposited by atomic layer deposition. Metallic source and drain contacts, separated by a 7 μm gap centred around the local gate, were patterned by optical lithography and e-beam evaporation of titanium (10 nm) and gold (60 nm).

**Graphene nanogaps**. A 600 nm layer of PMMA was spun onto CVD-grown graphene (Graphenea) on copper. The copper was subsequently etched in aqueous ammonium persulfate solution (3.6 g in 60 mL H$_2$O) overnight, transferred three times to fresh H$_2$O and scooped up using the substrate. Air bubbles were removed by partly submerging the sample in IPA. The sample was dried overnight and baked at 180 °C for 1 h prior to dissolving the PMMA in acetone at 50 °C for 3 h. Bow-tie shaped nanoribbons were patterned using e-beam lithography and the unexposed areas etched with O$_2$-plasma. The photoresist was subsequently removed using a flow of mr-REM 660 over the submerged sample, and subsequently soaking in fresh mr-REM 660 for 1 h and washed in acetone and IPA.

Graphene nanogaps were prepared by feedback-controlled electroburning of the nanoribbons until the resistance of the junction exceeded 500 MΩ. The empty nanogaps were characterised by measuring a current-map as a function of bias (±0.5 V) and gate voltage (±5 V) at 77 K in order to exclude devices containing residual graphene quantum dots[23]. Molecules were dropcast from solution (3 μM in toluene), and allowed to dry in air prior to performing measurements. Various stages of this procedure are illustrated in Supplementary Fig. 1.

We also fit the obtained *IV* characteristics (for graphene junctions prior to molecular deposition) to the (asymmetric) Simmons model[20,21,38]. The details of this procedure are described in the Supporting Information of Ref. [23]. Finally, we note that the edge-chemistry cannot be controlled during the process of electroburning. Controlling this aspect of electroburning constitutes a major technological challenge.

## Data availability
The data that support the findings of this study are available from the authors on reasonable request.

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

## Acknowledgements
We thank G. Holloway for developing the method to fabricate the 3-terminal electrode design. This work was supported by the EPSRC (grants EP/N017188/1 and EP/R029229/1). J.K.S. thanks the Clarendon Fund, Hertford College and EPSRC for financial support. E.M.G. acknowledges funding from the Royal Society of Edinburgh and the Scottish Government, J.A.M. acknowledges funding from the Royal Academy of Engineering.

## Author contributions
J.O.T., B.L., J.K.S., G.A.D.B., H.L.A. and J.A.M. guided the study. J.O.T. synthesised compound **M**. J.O.T. and B.L. performed graphene transfer and patterning, cryogenic measurements and data analysis. J.O.T. performed DFT calculations. B.L. and J.K.S. fitted the data. J.K.S. and E.M.G. developed the theory and fitting models. K.W. fabricated the substrates under supervision of J.B. All authors contributed to writing and editing the manuscript.

## Competing interests
The authors declare no competing interests.
