## [Peer Review File · Nature Communications]

Reviewers' comments:

Reviewer #1 (Remarks to the Author):

Thomas et al. report in "Understanding Electron Transfer on the Single-Molecule Level" Zn-porphyrin molecules deposited into graphene nanogaps on 300 nm thermally grown SiO₂. They try to replicate the experimental conductance maps with Marcus theory which they find does not accurately replicate their experimental results as the Marcus theory fails at low bias (Fig 4). Also, the Landauer model does not explain asymmetry in the I-V curves. This is indeed an important topic, and I find the concepts very time. The paper could be published after the following points have been addressed. The impact of the work is not clear, and I recommend to elaborate a bit more in the Introduction how their work compares to that done by others in different types of devices. The issue the authors try to address has more profound implications, and some terminology is a bit confusing. At the end of the introduction they refer to a semi-classical model by Nitzan and co-workers and not to the Marcus theory, but they do not state what that model is. The way this part is written is as if where the Marcus theory is not quantum corrected but of course that was done by Marcus himself. The authors should explain in more detail the model by Nitzan which combines elements from Landauer and Marcus theory to describe charge transfer in molecular junctions – this model has recently been experimentally verified in their reference 25 in a different type of junction. It is clear that different types of junctions sample different extremes and every model has advantages and disadvantages. A more balanced introduction would greatly help to define the impact of their work. In addition, Gerischer and Hopfield already pointed out decades ago the importance of connecting both Landauer and Marcus theories, and other attempts have been made recently (see for example the work of Bevan J. Chem. Phys. 146, 134106 (2017), J. Phys. Chem. C, 2016, 120 (1), pp 179–187 or J. Chem. Phys. 149, 104109 (2018). In the field of thin film electronics this issue has been also well recognised, see for example Phys. Rev. B 93, 140206(R) (2016) or Nat Commun. 4, 1710 (2013). So this topic has been studied from quite different directions and the authors could detail more how their findings fit in the big picture.

The energy level diagram in Fig 1d is quite generic, which molecular orbitals are involved, the HOMO/LUMO of the porphyrine moiety supposedly? Or what is meant with "charge transport through the junction as a sequence of reduction and oxidation reactions occurring", can the authors assign which orbital(s) is(are) involved? The discussion on page 4 is vague as charge transfer from and to the molecule is discussed, but it would be helpful to simply say explicitly which parts of the molecule are involved which the redox processes. The electrochemistry of porphyrins have been well-studied for decades and, for instance, deformation of the ring is important during oxidation: which vibrational modes are involved and how do the large substituents affect the redox properties (ie, reorganization energies) of the molecules they studied? In the SI they elaborate more on the calculations, but it is not clear how they obtained the results. I suggest to include the MO plots of the relevant orbitals of both the neutral and charged molecules involved in charge transport, and include the bulky side groups as they will affect reorganization energies. Electrochemical studies have shown before that the formal oxidation potentials can change significantly with steric hindrance and thus their calc values need to be checked.

The authors mention that their molecules pi-pi stack to the graphene electrode, but the molecules also contain long alkyl chains (as mentioned above) which also bind quite significantly to graphene which have not been drawn. The trihexylsilyl side groups are quite bulky and should also be drawn explicitly so the reader can have a balanced idea how these molecules really could bridge the graphene gap, in panel a these bulky features have been ignored. In the main text, it is not explained why the long alkyl chains and bulky THS groups are needed: why were such a bulky molecules used, is sterical hindrance not an issue hampering pi-pi stacking for instance (besides affecting reorganization energies)?

Figure S1d is very much appreciated, but this figure also shows that their devices are rather dirty and

the graphene around the gap seems to be different. These features are not explained but should be discussed. In their models/pictures they assume pristine graphene however this seems not to be the case, I would expect graphene oxides to be present. Graphene oxides on their own give rise to temperature effects in molecular junctions (see for a recent example J. Phys. Chem. C 2018, 122, 9731–9737). In Fig. S1e they show a fit to the Simmons equation but this is not further discussed, what were the fitting parameters used to obtain the fit (is this fit really useful)? Their λ_0 is very small (26 meV): can it be assigned with great certainty? In how many devices the effects have been seen (what is their experimental reproducibility)?

On page S17, they should use the static dielectric constant of HfO₂, not SiO₂

Spelling error second line page S23 “junctions as the liquid-nitrogen temperature” should be “junctions at the liquid-nitrogen temperature”

Quite a few other binding groups were tested, but this is very briefly mentioned in the main text. The authors could elaborate much more on these findings and how these support their model in main text.

The last paragraph on page 9 is quite generic, a bit over the top, and is out of place. I suggest to remove it.

Reviewer #2 (Remarks to the Author):

The authors present a theory on electron transfer in single-molecule junctions and compare the theoretical predictions with experimental data. The paper is written as if the description is very general and the observed features should be present in all molecular junctions. I do find the approach nice but I am bit confused by two aspects:

(i) I am not sure to what extent the theory is new; nuclear tunnelling has been considered before in for instance a Franck-Condon description of the coupling of transport to vibrations. In this respect, the authors do not make clear enough what the new aspects are regarding their previous work and the work by others (e.g. Nanoletters 5 (2005) 125-130). As far as I understand it, the incorporation of the back ground seems to be the new element but this on its own would not warrant publication in Nature Communication. In addition, the assumptions on the phonon contribution to the background are rather vague as is the effect of charge state of the molecule on the it.

(ii) In the comparison with the data, there are several aspects that should be cleared up. First of all, the lines in the stability diagram (Fig. 1c) do not run parallel to the Coulomb edges. This may indicate that they originate from states in the graphene, which would mean that the environment and the cause of the background tunneling can just be due to the use of graphene leads and that when using e.g. gold electrodes the background tunnelling description may not be needed for a proper description of transport. Furthermore, the authors should show (maybe in the SI) the whole stability diagram as a reference (this should be common practice). Second, in comparing the calculated and measured stability diagram the authors state that excellent agreement is found. I do not find that statement very convincing as reproducing the diamond is not a very accurate measure. More insight should be given in the assumptions and fit parameters behind the fit considering the number of free parameters.

In summary, I see the presented theory as a possible explanation for their data but I am not so sure that it is widely applicable to other systems and that other (related) models may not offer an equally good description of transport. I therefore cannot recommend publication in Nature Communications. Some other points:

1. I find the paper sometimes difficult to read as it contains very long paragraphs; the presentation of the work could thus be improved.
2. In the paper electron transfer is used as the term for electrons to be transported from one electrode to the other. Traditionally electron transfer is used for contactless measurements when

irradiating molecules with e.g. light and studying the transfer of charges within the molecule. Electron transport is used when electrodes are involved and charges move through the molecule. Is there a specific reason for using electron transfer in this case?

3. The kink in the measured Coulomb diamond in Fig. 2b seem to indicate the presence of a second dot. Can the model be applied to such a situation without implicitly taking this second dot into account?

4. On page 9 the authors write "Quantum mechanically, however,". This sounds exactly as the Franck-Condon model (see also above). Is this what is meant here?

5. The gate voltage range in Fig. 4 is different than for the plot in Fig. 1 (due to a different gate geometry and fabrication technology). It is not clear to me if the results of the two sample lay-outs be compared just like that especially concerning the background modelling.

6. An additional question on Fig. 4 concerns the non-closing of some of the diamonds. What is the cause of this?

Reviewer #3 (Remarks to the Author):

The authors study dependence of current through porphyrine molecule noncovalently attached to graphene electrodes on bias voltage, gate voltage, temperature, and anchor group. Experimental data are fitted by theoretical model assuming single electronic level for the bridging molecule, single quantum harmonic vibrational mode coupled to that level, other level-shifting interactions via empirical spectral density, rate theory for electron exchange with electrodes, no other dynamic variables and no other specific interactions with electrodes and environment. The results are interesting and can be published as is. However, judging by novelty and significance, Nature Communications might be inappropriate journal, because I do not see how this paper can influence thinking in the field. The problem is that claims made in the manuscript are not novel but rather elucidate known facts (from reviews and textbooks) using state of the art experimentation and modeling. Let me analyze statements made in the abstract to see what can be a major claim of this work:

1) "We observe a simultaneous breakdown of quantum coherent Landauer and semi-classical Marcus theory."

The bridging molecule is large and flexible and the contacts are weak, so that no ballistic transport is expected according, e.g., to p.20 of Ref.16. At the same time semi-classical Marcus theory is always broken-down at low enough temperature according to Section 6.4.3 of the book [V May, O Kuhn, Charge and energy transfer dynamics in molecular systems (Wiley, 2004)], and for pi-conjugated molecules this critical temperature is lower than their thermal decomposition temperature.

2) "We propose a theoretical model based on a quantum master equation,..."

All ingredients of the proposed model are known (they are properly cited in the current work).

3) "... and demonstrate that it quantitatively describes rates of electron transfer in single molecules."

Because the model is empirical (fitted to experiment), it is designed to be quantitatively good.

4) "We show that nuclear tunnelling enhances the rates of low-energy electron transfer,..."

This the consequence of use of the correct phonon correlator in Eq.5 instead of a priori incorrect high-

temperature replacement of $\coth(x)$ by $1/x$.

5) "... and demonstrate that the rates are sensitive to both the outer and inner-sphere environmental interactions."

This is the most common situation already recognized in the review [R A Marcus, Chemical and electrochemical electron-transfer theory, *Annu Rev Phys Chem* 15, 155 (1964)]

6) "We find that the nuclear dynamics accompanying electron transfer must be treated quantum mechanically as the quantitative validity of Marcus theory is expected to occur at temperatures exceeding 298K."

See the above discussions.

To summarize, all the statements made by the authors are correct and well supported in this work but not novel enough to influence thinking in the field.

There is also one comment concerning DFT calculations: low-frequency modes in organic molecular systems are usually substantially anharmonic (with nonlinear vibronic coupling) and intermix with other low-frequency intra- and intermolecular modes. For this reason the assignment (on p.3) of 4.9 meV vibronic progression with 6.0 meV mode calculated for a molecular fragment at some point of its potential energy surface sounds statistically meaningless.

Reply to Reviewers' Comments and Details of Revisions

Reviewer #1 (Remarks to the Author):

Comment 1: Thomas et al. report in "Understanding Electron Transfer on the Single-Molecule Level" Zn-porphyrin molecules deposited into graphene nanogaps on 300 nm thermally grown SiO₂. They try to replicate the experimental conductance maps with Marcus theory which they find does not accurately replicate their experimental results as the Marcus theory fails at low bias (Fig 4). Also, the Landauer model does not explain asymmetry in the I-V curves. This is indeed an important topic, and I find the concepts very time. The paper could be published after the following points have been addressed.

Reply 1: *We are glad that the Reviewer finds the subject of our work to be important and timely, and that they recommend publication of a revised manuscript in Nature Communications. We thank them for their thorough review of our work and welcome their suggestions.*

Comment 2: The impact of the work is not clear, and I recommend to elaborate a bit more in the Introduction how their work compares to that done by others in different types of devices. The issue the authors try to address has more profound implications, and some terminology is a bit confusing. At the end of the introduction they refer to a semi-classical model by Nitzan and co-workers and not to the Marcus theory, but they do not state what that model is. The way this part is written is as if where the Marcus theory is not quantum corrected but of course that was done by Marcus himself. The authors should explain in more detail the model by Nitzan which combines elements from Landauer and Marcus theory to describe charge transfer in molecular junctions – this model has recently been experimentally verified in their reference 25 in a different type of junction. It is clear that different types of junctions sample different extremes and every model has advantages and disadvantages. A more balanced introduction would greatly help to define the impact of their work. In addition, Gerischer and Hopfield already pointed out decades ago the importance of connecting both Landauer and Marcus theories, and other attempts have been made recently (see for example the work of Bevan J. Chem. Phys. 146, 134106 (2017), J. Phys. Chem. C, 2016, 120 (1), pp 179–187 or J. Chem. Phys. 149, 104109 (2018). In the field of thin film electronics this issue has been also well recognised, see for example Phys. Rev. B 93, 140206(R) (2016) or Nat Commun. 4, 1710 (2013). So this topic has been studied from quite different directions and the authors could detail more how their findings fit in the big picture.

Reply 2: *We thank the Reviewer for their recommendations. We agree that several issues here indeed require clarification.*

The first point raised by the Reviewer concerns the application of Marcus theory in the transport setting considered here. Conventional Marcus theory has been developed to describe (non-adiabatic) electron transfer within a donor-acceptor system. A generalisation of this theory to describe electron transfer between a metallic electrode and molecular species is commonly attributed to Chidsey (Chidsey C., Science, 1991). That approach (sometimes referred to as Marcus-Hush-Chidsey theory) has been applied to transport through molecular junctions first by Ulstrup, Kuznetsov and coworkers (beginning in the late 90s), and more recently by Nitzan et al. This description has indeed been recently verified in Ref. 25, however in a different type of junction (a self-assembled monolayer junction) and at significantly higher temperatures than in our work. Conversely, we demonstrate the

failure of this classical (Marcus) description of transport (including the above-mentioned generalisations).

Secondly, as the Reviewer correctly points out, the classical character of Marcus theory has long been well understood and both semi-classical and fully quantum-mechanical theories of electron transport have been formulated in the past (also by Rudolph Marcus himself). The importance of nuclear tunnelling in electron transfer has been therefore repeatedly verified beginning with the work of Miller et al. (J. R. Miller et al. JACS 106.18 (1984): 5057-5068). As we discuss in the manuscript, here, we assess the importance of nuclear tunnelling in the steady-state transport setting and on a single-molecule level.

In our work, we show that none of the most commonly used theoretical models are sufficient to describe the experimentally-observed behaviour in our devices. In particular, we observe a breakdown of the non-interacting Landauer and the aforementioned semi-classical Marcus theory as well as the single-mode Franck-Condon model. Instead, we demonstrate that the resonant charge transport through weakly-coupled single-molecule junctions can only be understood as a sequence of non-adiabatic electron transfers, and described using a quantum-master-equation-based theoretical model.

The abstract and the introduction of the manuscript have been re-written in the light of the above comments. We hope the revised version clarifies how our approach connects to the prior literature and its advance over previous work.

Comment 3: The energy level diagram in Fig 1d is quite generic, which molecular orbitals are involved, the HOMO/LUMO of the porphyrine moiety supposedly?

Reply 3: *The entire stability diagram of device A is now in the SI (Section S6), and we observe two charge transitions. As graphene on HfO₂ (and indeed SiO₂) is p-doped, the molecular anchor groups are electron-rich, and the transitions appearance at negative gate voltages, we assign these transitions to oxidation processes, i.e. transitions between the (electronic ground states of) N/N-1 and N-1/N-2 charge states where the N-charge state corresponds to the neutral molecular species. It is the N-1/N-2 transition that we study in detail in Fig 1/ Fig 2. The device became too unstable after the temperature-dependent measurements to study the N/N-1 transition in detail. With this in mind, we expect that the two relevant orbitals are the SOMO of the N-1 charge state (from which an electron is removed for oxidation to N-2), and the LUMO of the N-2 state (to which an electron is added for reduction to N-1).*

The assignment of the transition is now stated in the manuscript in the following text (Results and Discussion, page 3):

The transition observed here is most likely the transition between the N-1 and N-2 charge states (where N charge state corresponds to the neutral molecular species, i.e. the M⁺/M²⁺ transition), see SI for assignment and frontier orbitals of these charge states.

The entire stability diagram added to the revised SI has been assigned, Section S6, Figure S15. We have also modified Fig 1d, to display the M²⁺ to M⁺, above the energy level diagram. Finally, we have plotted the MOs of the relevant orbitals in Figure S3.

We also assign the charge transition in devices B-D as the N-1/N transition because, as above, we expect the oxidation potential of the electron-rich molecule to most closely match the Fermi level of the p-doped graphene. We have added the following text to state this (Results and Discussion, page 8):

From:

The frequencies and coupling strengths of the molecular modes were calculated using DFT and correspond to an inner-sphere reorganisation energy of $\lambda_i = \hbar \sum_q \omega_q S_q$ of 67 meV (see SI for details) whereas the background is again modelled as a structureless super-Ohmic spectral density.

To:

The frequencies and coupling strengths of the molecular modes were calculated using DFT and correspond to an inner-sphere reorganisation energy of $\lambda_i = \hbar \sum_q \omega_q S_q$ of 67 meV for the N/N-1 transition, which we consider the most likely assignment (see SI for details of the calculation). The background is again modelled phenomenologically as a structureless super-Ohmic spectral density.

Comment 4: Or what is meant with “charge transport through the junction as a sequence of reduction and oxidation reactions occurring”, can the authors assign which orbital(s) is(are) involved?

Reply 4: As discussed in the manuscript and schematically shown in Fig 1, we describe the overall charge transport as a series of electron transfers (oxidations and reductions). We agree that referring to those processes as ‘reactions’ might have been confusing.

The manuscript has been revised accordingly in Results and Discussion, page 4:

From:

In order to account for the observed behaviour, we describe the charge transport through the junction as a sequence of reduction and oxidation reactions occurring at the source and drain electrodes. The rates of these reactions depend on the interaction between the electronic and vibrational degrees of freedom of the molecule and its environment as illustrated in Figure 1d.

To:

In order to account for the observed behaviour, we describe the overall charge transport through the junction as a sequence of electron transfers occurring at the source and drain electrodes, as schematically shown in Figure 1a. The rates of these oxidation and reduction processes depend on the interaction between the electronic and vibrational degrees of freedom of the molecule and its environment as illustrated in Figure 1d.

Reply 3 addresses the issue of which orbitals are involved (shown in Section S2, Figure S3).

Comment 5: The discussion on page 4 is vague as charge transfer from and to the molecule is discussed, but it would be helpful to simply say explicitly which parts of the molecule are involved which the redox processes. The electrochemistry of porphyrins have been well-studied for decades and, for instance, deformation of the ring is important during oxidation: which vibrational modes are involved and how do the large substituents affect the redox properties (ie, reorganization energies) of the molecules they studied? In the SI they elaborate more on the calculations, but it is not clear how they obtained the results. I suggest to include the MO plots of the relevant orbitals of both the neutral and charged molecules involved in charge transport, and include the bulky side groups as they will affect reorganization energies. Electrochemical studies have shown before that the formal

oxidation potentials can change significantly with steric hindrance and thus their calc values need to be checked.

Reply 5: *The additional charge density is localised/withdrawn predominantly from the porphyrin ring and the anchor groups. This is clearly demonstrated in the revised version of the SI (Section S2, Figure S3) where we have added the requested plots of the HOMO, SOMO and LUMO orbitals obtained from DFT calculations for the relevant charge states.*

The calculations of the MOs included 3,5-bis(trimethylsilyl)phenyl groups to demonstrate that very little electron density is localised on them. This shows that adding in the full 3,5-bis(trihexylsilyl)phenyl groups, which would greatly increase computational cost, would not offer any extra value to the work. Furthermore, as the frontier orbitals have very little electron density on the side groups the charging of the molecule is not expected to alter the equilibrium nuclear positions of atoms in these bulky side-groups. Therefore we feel this justifies the substitution of these side-groups for hydrogens in the full calculation of the inner sphere reorganisation energy and vibrational couplings.

We certainly agree however that these large groups will affect the outer-sphere reorganisation energy (and molecule-lead coupling) by influencing the alignment of the molecular structure within the junction. Such an effect will vary from device -to-device and is conveniently captured by the magnitude of the background spectral density, which is one of the big selling points of our work.

We have now addressed these points in the manuscript by adding the following sentences to Results and Discussion, page 3:

3,5-Bis(trihexylsilyl)phenyl aryl groups increase the solubility of the porphyrin and prevent aggregation, however, we do not expect them to directly contribute to the charge transport, as DFT calculations indicate that during the oxidation/reduction of the molecular species the additional charge density is localised on the porphyrin ring and the anchor groups (although the aryl groups may affect the molecule-lead and outer sphere electronic-vibrational coupling).

and in a new subsection (Section 4.4) in the revised version of the SI. We have also plotted some examples of the strongly-coupled vibrational modes in the SI (Section S4, Figure S12) which correspond to (i) a collective twisting of the anchor group [a low-frequency mode at ~ 6 meV] and (ii) a higher-frequency breathing/stretching mode of the porphyrin ring [~ 175 meV].

We note however that the exact nature of these vibrational modes is inconsequential to the behaviour of the studied system and the conclusions of this work.

Finally, we note that while these groups can also affect the oxidation/reduction potentials, the position of the Coulomb peaks in our study is an empirical parameter. The effect mentioned by the Reviewer will therefore have no bearing on the conclusions of this work.

Comment 6: The authors mention that their molecules pi-pi stack to the graphene electrode, but the molecules also contain long alkyl chains (as mentioned above) which also bind quite significantly to graphene which have not been drawn. The trihexylsilyl side groups are quite bulky and should also be drawn explicitly so the reader can have a balanced idea how these molecules really could bridge the graphene gap, in panel a these bulky features have been ignored.

Reply 6: *The pyrene-based anchor group contains long alkyl chains because in a previous study (Limburg et al. Adv. Funct. Mater. 2018, 28, 1803629) we showed that they had the effect of*

increasing the probability of forming a molecular junction when compared to a simple pyrene anchor. Presumably this is due to the binding of the alkyl chains on the graphene, which is significant (770 meV per $-C_{12}H_{25}$ chain), as the reviewer points out. We have adjusted the text (Results and Discussion, p3) to make the reader aware that these interactions will be present (in addition to π - π stacking) and added the reference to the paper focussed on the design of these molecules:

From:

Zinc porphyrin molecules, functionalised with anchor groups designed to bind to the graphene electrodes via π - π stacking (Figure 1b), are deposited from solution.

To:

Zinc porphyrin molecules, functionalised with anchor groups that have been designed to bind to the graphene electrodes via π - π stacking and van der Waals interactions¹⁹ (Figure 1b), are deposited from solution.

The energetically accessible electronic states have little-to-no density on these non-conjugated groups. Therefore they should only affect the charge transport properties in a similar way to the side groups, by modifying the way the molecule binds within the junction, and consequently affecting the molecule-lead coupling strength and the outer sphere reorganisation. The details of these interactions will vary from device-to-device and the molecule-lead coupling strength and the outer sphere reorganisation are already free parameters within our model. Therefore, for clarity, we have decided to include only the redox-active parts of the molecule in Figure 1(a). However we have explicitly stated this by adding the following text (Figure 1 caption):

For clarity, the bulky side-groups are not shown.

Comment 7: In the main text, it is not explained why the long alkyl chains and bulky THS groups are needed: why were such a bulky molecules used, is sterical hindrance not an issue hampering pi-pi stacking for instance (besides affecting reorganization energies)?

Reply 7: *The bulky THS groups were used to increase the solubility of the porphyrin species and prevent molecular aggregation (see response to Comment 5). Empirically, we do not find this sterical hindrance to hamper the efficiency of pi-pi stacking, as explained in our previous work (Limburg et al. Adv. Funct. Mater. 2018, 28, 1803629). As mentioned in our response to Comment 6, the alkyl chains on the anchor increase the probability of forming a molecular junction.*

We agree that the molecular design needed to be clarified. We believe the additional text in the revised version of the manuscript (detailed in response to Comment 5), and the reference to our previous work in which molecular design is discussed (Limburg et al. Adv. Funct. Mater. 2018, 28, 1803629), resolves this issue.

Comment 8: Figure S1d is very much appreciated, but this figure also shows that their devices are rather dirty and the graphene around the gap seems to be different. These features are not explained but should be discussed. In their models/pictures they assume pristine graphene however this seems not to be the case, I would expect graphene oxides to be present. Graphene oxides on their own give rise to temperature effects in molecular junctions (see for a recent example J. Phys. Chem. C 2018, 122, 9731–9737).

Reply 8: *The aforementioned features originate during the process of electroburning: due to the high temperature (occurring in the central region during the electroburning procedure) contamination in the central region is removed. This has been confirmed by comparing AFM images of junctions before*

and after electroburning, shown in the SI of our previous work (Limburg et al. *Adv. Funct. Mater.* 2018, 28, 1803629). We have modified the text in the caption of Figure S1 to clarify this:

From:

A cleaning circle is observed due to Joule heating during the process.

To:

A cleaning circle is observed due to Joule heating during the process: high temperature occurring in the central region during the electroburning procedure effectively removes the contamination.

Secondly, the reviewer is correct that we cannot control the edge chemistry of graphene. While small amount of graphene oxide may be present, the results show that the graphene electrodes still behave as metallic. This can be inferred, for instance, from the fact that before the molecular deposition the graphene junction can be very well described using the Simmons model (see below).

This is now discussed and clarified in the revised version of the SI (Section S1) in the following text:

Finally, we note that the edge chemistry cannot be controlled during the process of electroburning. Controlling this aspect of electroburning constitutes a major technological challenge.

Comment 9: In Fig. S1e they show a fit to the Simmons equation but this is not further discussed, what were the fitting parameters used to obtain the fit (is this fit really useful)?

Reply 9: We use the Simmons fit to confirm that the fabricated device (after the process of electroburning and before the molecular deposition) constitutes a tunnel junction. The fitting parameters are the barrier height, a barrier asymmetry, the barrier width, and a prefactor. We agree with the Reviewer that the fitted parameters are not necessarily reliable (and hence are not reported) but it is good to show that our experimental results can be fitted with a reasonable model for electron tunnelling through vacuum.

The parameters used to obtain this fit as well as the points raised above are now briefly discussed in the revised version of the SI (Section S1), and the interested reader is directed to references describing the procedure in detail.

We also fit the obtained IV characteristics (for graphene junctions prior to molecular deposition) to the (asymmetric) Simmons model.²⁻⁴ The details of this procedure are described in the Supporting Information of Ref. 1.

Comment 10: Their λ_0 is very small (26 meV): can it be assigned with great certainty? In how many devices the effects have been seen (what is their experimental reproducibility)?

Reply 10: Device A has been studied only at relatively low bias (± 50 meV). Consequently, it was only possible to study the low-frequency part of the vibrational environment. The outer-sphere reorganisation energy λ_o corresponds only to the relevant (low-frequency) part of the vibrational environment and therefore provides a lower bound for λ_o , hence it being a small number. For the same reason, we can consider only a single mode for the inner-sphere reorganisation energy, even though at higher bias voltages we should clearly address many more (as we do in the other devices discussed in the manuscript, which have been measured at much higher voltages). This issue is now clarified in the revised version of the manuscript by the addition of the following text (Results and Discussion, page 6):

From:

From the fit we obtain $\lambda_o = 26$ meV and $\hbar\langle\omega_o\rangle = 25$ meV.

To:

From the fit we obtain $\lambda_o = 26$ meV and $\hbar\langle\omega_o\rangle = 25$ meV (we note that only the low-frequency part of the outer-sphere background can be extracted from the low-bias measurements considered here).

λ_o is not a parameter that is reproducible between devices -for the devices where we measure over a larger bias range (B-D) and (F-N) we obtain λ_o values in the range of 110–250meV. As with the molecule-electrode coupling, the device-to-device variability in λ_o can be attributed to small variations in the distance of the porphyrin from the dielectric surface. We find a vibrational mode at ~ 5 meV (and therefore the same low-frequency λ_i) for both devices **A** and **E**.

Comment 11: On page S17, they should use the static dielectric constant of HfO₂, not SiO₂

Reply 11: Only device **A** (and **E**, in the SI) was fabricated on a HfO₂ substrate. On the other hand, the remaining devices, for which the calculation of molecule-substrate distance was carried out (**B-D**, **F-N**) were fabricated on an SiO₂ substrate. The use of a SiO₂ substrate was mentioned in the main text (Results and Discussion, page 8), however we have updated the following text to make sure this point is not overlooked (Results and Discussion, page 10):

From:

we estimate that the above values of λ_o correspond to the porphyrin molecules being roughly up to 0.71, 0.51 and 0.58 nanometres away from the dielectric substrate

To:

we estimate that the above values of λ_o correspond to the porphyrin molecules being roughly up to 0.71, 0.51 and 0.58 nanometres away from the SiO₂ dielectric substrate

Comment 12: Spelling error second line page S23 “junctions as the liquid-nitrogen temperature” should be “junctions at the liquid-nitrogen temperature”

Reply 12: We thank the Reviewer for pointing this out. This typo is fixed in the revised version of the SI.

Comment 13: Quite a few other binding groups were tested, but this is very briefly mentioned in the main text. The authors could elaborate much more on these findings and how these support their model in main text.

Reply 13: We thank the Reviewer for this suggestion. These findings are discussed in more detail in the revised version of the manuscript by modifying the following text (Results and Discussion, page 11/12).

From:

In addition, in the SI we present data for 9 additional devices, comprising porphyrin molecules that differ only by the π -anchoring group. The values of λ_o obtained for these devices are in the range of 110 – 250 meV.

To:

In addition, in the SI we present data for 9 additional devices, comprising porphyrin molecules that differ only by the π -anchoring group. As for devices **B-D**, we calculate λ_i for the molecular species, and then fit our data to three parameters: Γ_S , Γ_D and λ_o . The results (presented in the SI) show the transport behaviour of these devices can also be successfully explained using our QME model (and not using the semi-classical Marcus theory). The values of λ_o obtained for these devices are in the range of 110 – 250 meV. As expected, the effectiveness of the theoretical approach used here does not depend on the exact chemical structure of the considered molecular species.

Comment 14: The last paragraph on page 9 is quite generic, a bit over the top, and is out of place. I suggest to remove it.

Reply 14: *We thank the Reviewer for this suggestion. The paragraph has now been removed.*

Reviewer #2 (Remarks to the Author):

Comment 1: The authors present a theory on electron transfer in single-molecule junctions and compare the theoretical predictions with experimental data. The paper is written as if the description is very general and the observed features should be present in all molecular junctions. I do find the approach nice but I am bit confused by two aspects:

Reply 1: *We thank the Reviewer for their valuable comments and positive remark about our approach. We hope that our reply below (and changes made to the manuscript) will clarify the issues raised by the Reviewer.*

Comment 2: (i) I am not sure to what extent the theory is new; nuclear tunnelling has been considered before in for instance a Franck-Condon description of the coupling of transport to vibrations. In this respect, the authors do not make clear enough what the new aspects are regarding their previous work and the work by others (e.g. Nanoletters 5 (2005) 125-130). As far as I understand it, the incorporation of the back ground seems to be the new element but this on its own would not warrant publication in Nature Communication.

Reply 2: *The Reviewer correctly points out that the effects of nuclear tunnelling are captured by the Franck-Condon model (for instance in the form introduced by Koch and von Oppen in 2004). The Reviewer is further correct in noting that the background (outer-sphere) contribution is typically ignored when modelling charge transport through molecular junctions: a shortcoming we rectify in our work.*

However, this is not the only novel element of our work. As we now more clearly discuss in the revised version of the manuscript, our investigations have for the first time given us a full understanding of resonant charge transport in graphene-based molecular junctions. We have demonstrated that the charge transport in this regime is governed by an interplay between electron-electron interactions and electron-vibrational interactions of the inner-sphere as well as the outer-sphere origin. We show this by quantitatively describing the electron transport through porphyrin junctions (at 77 K) with only 3 free parameters.

We have therefore shown that, in order to understand the experimental behaviour, significantly more sophisticated modelling (than what is usually done) is required.

In fact, the Franck-Condon model mentioned by the Reviewer can be obtained as one of the limits of our theoretical approach provided that we:

- Consider only a single molecular vibrational mode (and ignore the outer-sphere background)
- Ignore the effects of electron-electron interactions
- Disregard lifetime broadening

In summary, our theoretical model is able to simultaneously account for the effects of outer- and inner-sphere vibrational coupling, the effects of electron-electron interactions and lifetime broadening. While doing so, we still limit the number of variables to the same number as present in other models. Our fitting parameters – the tunnel coupling to source and drain and the outer-sphere reorganisation energy – are the only parameters that cannot be derived from first principles due to the random orientation of the molecule in the junction. The ability to obtain a quantitatively good fit for the experimental transport data should not be taken lightly. As we show in the SI, other mentioned models, which can all be derived as limits of our model, do not show a quantitatively or even qualitatively correct fit.

The abstract, introduction and conclusion of our manuscript were re-written considering above comments.

Comment 3: In addition, the assumptions on the phonon contribution to the background are rather vague as is the effect of charge state of the molecule on the it.

Reply 3: *The vibrational background is modelled in a phenomenological fashion using a generic super-ohmic spectral density given explicitly in the SI (equation S19). This choice is motivated by the fact that the super-ohmic spectral density is usually the appropriate description of (deformation) vibrational interactions in semiconductor quantum dots [see for instance T. Calarco et al. Phys. Rev A 68, 012310 (2003)], and has been widely used in theoretical description of charge and energy transport [see for instance S. Jang et al. New J. Phys. 15, 10 (2013): 105020].*

We assume that the phononic spectral density has the same shape regardless of the considered charge transition, however its contribution to the overall reorganisation energy might differ, which is captured by the model.

We have clarified this by modifying the following text and additional references in the main text (Results and Discussion, page 6):

From:

However, a spectral density consisting of only this single mode cannot fully reproduce the experimental data. Only if we include a structureless super-Ohmic background²² accounting for the coupling to the substrate, do we find a good agreement with the empirical data, as shown in Figure 2a (top panel)

To:

However, a spectral density consisting of only this single mode (the usual Franck-Condon model)⁴ cannot fully reproduce the experimental data. Only if we account for the coupling to the substrate, do we find a good agreement with the empirical data, as shown in Figure 2a (top panel). We model this outer-sphere background using a structureless super-Ohmic spectral density^{26,27} which constitutes the simplest (phenomenological) description of this environmental contribution, see SI.

Comment 4: (ii) In the comparison with the data, there are several aspects that should be cleared up. First of all, the lines in the stability diagram (Fig. 1c) do not run parallel to the Coulomb edges. This may indicate that they originate from states in the graphene, which would mean that the

environment and the cause of the background tunneling can just be due to the use of graphene leads and that when using e.g. gold electrodes the background tunnelling description may not be needed for a proper description of transport.

Reply 4: As we have previously demonstrated, density of states (DOS) fluctuations can indeed result in conductance peaks in resonant transport through molecular junctions (see our previous work Gehring et al. ACS Nano 2017, 11 (6), pp 5325–5331). As correctly pointed out by the Reviewer, such features are characterised by lines running not parallel to the edges of the Coulomb diamond.

Conversely, we argue that the features observed here do not stem from DOS fluctuations. We agree that it may be generally difficult to differentiate between the parallel and non-parallel lines. That is why the plot in top half of Fig. 1(c) showing the conductance peaks (corresponding to vibrational excitations) was obtained by averaging the conductance in a way which should average out the density of states fluctuations. In addition, a larger stability diagram obtained at 7 K shows more clearly that the lines do run parallel, and is now shown in the SI (Section S6, Figure S15).

Furthermore, DOS fluctuations cannot explain the overall shape of the IV characteristics (linear-like increase in current over few hundred meVs of applied bias voltage) or the asymmetry of the presented stability diagrams (which stems from the presence of electron-vibrational interactions and is very well captured by our model).

We have now clarified this point in the revised version of the manuscript (Results and Discussion, page 3) by adding the following sentences and reference:

We note that density of states fluctuations in the leads cannot explain the observed behaviour. Density of state fluctuations give rise to lines non-parallel to the edges of the Coulomb diamond.²¹ The features observed here are on the other hand parallel to the aforementioned edges (as is more clearly visible in Figure S15 in the SI).

Comment 5: Furthermore, the authors should show (maybe in the SI) the whole stability diagram as a reference (this should be common practice).

Reply 5: As per Reviewer's suggestion, the whole stability diagram is now attached in the revised version of the SI (Section S6, Figure S15).

Comment 6: Second, in comparing the calculated and measured stability diagram the authors state that excellent agreement is found. I do not find that statement very convincing as reproducing the diamond is not a very accurate measure. More insight should be given in the assumptions and fit parameters behind the fit considering the number of free parameters.

Reply 6: We agree with the Reviewer that reproducing the diamond (the region of Coulomb-blockade) is not a very accurate measure for comparing the calculated and measured stability diagram. Furthermore, in general, differences in colorscale plots are more difficult to perceive than for line-graphs. That is why in Figure 2a we demonstrate our approach gives excellent agreement between calculated and measured resonant IV traces when compared to Landauer (no coupling) or Franck-Condon (single-mode, no background) models. Furthermore, we add experimental and theoretical gate traces at multiple bias voltages to Figure 4 for device C. In the revised version of the SI (Section S8, Figures S18, S19), we have added additional experimental and theoretical gate traces at multiple bias voltages for B and D. These traces clearly demonstrate the excellent agreement between the calculated and measured values of electric current for our QME model.

As we discuss in the manuscript, the theoretical fit used for experimental data measured at 77 K features only 3 free (fitting) parameters: two molecule-lead coupling strengths and the background reorganisation energy (all three of which will depend on the molecular alignment within the junction and must vary from device to device). We note that the same number of free parameters is present within the conventional Marcus theory while the single-mode Franck-Condon model would typically feature four free parameters.

Comment 7: In summary, I see the presented theory as a possible explanation for their data but I am not so sure that it is widely applicable to other systems and that other (related) models may not offer an equally good description of transport. I therefore cannot recommend publication in Nature Communications.

Reply 7: *We hope the clarifications we have provided convince the Reviewer that our theory presents the correct explanation of our data, for all presented devices. As mentioned in Reply 6 we have presented a model that accounts for more phenomena (lifetime broadening, electron-electron interactions, inner and outer sphere reorganisation) than alternative models, without the need for additional parameters, that accurately reproduces experimental data over large energy and a temperature range. Furthermore, as mentioned above, our approach reduces to a number of well-established models in various extreme limits. Therefore, since those reduced models have been applied to various junctions in the past, we strongly believe that our more general, and complete, model will also be applicable to other single-molecule-junction systems.*

We trust that publication of this manuscript in Nature Communications will give our approach the necessary exposure to see it adopted (and if required adjusted) for use with different systems and on different experimental platforms.

Comment 8: Some other points:

1. I find the paper sometimes difficult to read as it contains very long paragraphs; the presentation of the work could thus be improved.

Reply 8: *The paper has been re-structured in the light of the above comments.*

Comment 9:

2. In the paper electron transfer is used as the term for electrons to be transported from one electrode to the other. Traditionally electron transfer is used for contactless measurements when irradiating molecules with e.g. light and studying the transfer of charges within the molecule. Electron transport is used when electrodes are involved and charges move through the molecule. Is there a specific reason for using electron transfer in this case?

Reply 9: *In this work, we model the overall electron transport as a series of electron transfers (from the source electrode onto the molecule, and the molecular species into the drain electrode). Such description is justifiable in the case of weak molecule-lead coupling, as is the case here.*

We have modified the following sentences to hopefully clarify this reference to electron transfer in the context of charge transport (Results and Discussion, page 4):

From:

In order to account for the observed behaviour, we describe the charge transport through the junction as a sequence of reduction and oxidation reactions occurring at the source and drain electrodes.

To:

In order to account for the observed behaviour, we describe the overall charge transport through the junction as a sequence of **electron transfers** occurring at the source and drain electrodes, **as schematically shown in Figure 1a.**

Comment 10:

3. The kink in the measured Coulomb diamond in Fig. 2b seem to indicate the presence of a second dot. Can the model be applied to such a situation without implicitly taking this second dot into account?

Reply 10: *As the Reviewer correctly points out, the presence of the kink in Fig. 2b may indicate a presence of another quantum dot which does not directly contribute to the transport characteristics but rather capacitively couples to the molecular structure studied here [in an analogy to what has been observed in for example Appl. Phys. Lett. 104 233503 (2014) and Appl. Phys. Lett. 96 042114 (2010)]. The fact that the Coulomb diamond closes proves that the studied junction does not comprise two quantum dots in series. Therefore, our model (where only one quantum dot is considered) remains valid. The simple electrostatic effect of a capacitively-coupled nearby dot is naturally not accounted for in our model, neither does it affect the conclusions of our study.*

This issue is now briefly discussed, along with a reference, in the following text (Figure 2 caption):

At higher (positive) bias we observe switching in the stability diagram most likely resulting from a nearby charge trap.²⁸ This effect is however inconsequential to the phenomena discussed here.

Comment 11:

4. On page 9 the authors write “Quantum mechanically, however, ...”. This sounds exactly as the Franck-Condon model (see also above). Is this what is meant here?

Reply 11: *We refer there to the breakdown of the classical Marcus approach. As the Reviewer correctly points out, nuclear tunnelling is captured by the so-called Franck-Condon model. In our work, however, we go beyond this, usually single-mode case, and incorporate coupling to both the molecular and background vibrational environment, account for the effects of electron-electron repulsion, and capture the effects of lifetime broadening [c.f. Koch and von Oppen, PRL 94, 206804 (2004)] in order to correctly describe the data. As mentioned above, the Franck-Condon model (of Koch and von Oppen) can be obtained as one of the limits of our theoretical approach [see our **Reply 2**].*

We hope the following two sentences that have been added to the manuscript outline the differences between the Franck-Condon model and our own (Introduction, page 2):

Typically, resonant charge transport through molecular junctions is described using a single- or many-mode Franck-Condon model.^{4,5} While sometimes successful on a qualitative level,^{6,7} these approaches usually do not account for the broader environmental vibrational coupling and ignore the effects of electron-electron interactions.

Comment 12: 5. The gate voltage range in Fig. 4 is different than for the plot in Fig. 1 (due to a different gate geometry and fabrication technology). It is not clear to me if the results of the two sample lay-outs be compared just like that especially concerning the background modelling.

Reply 12: As the Reviewer correctly points out, the devices shown in Figures 1 and 4 were fabricated on different substrates and in slightly different device geometry. Nonetheless, since our modelling of the background environment is based on a generic super-ohmic spectral density, changes in the electrostatic environment can be conveniently captured by changes in the background reorganisation energy. We believe that the conclusions of this work are not affected by these small variations in the device geometry.

Comment 13: 6. An additional question on Fig. 4 concerns the non-closing of some of the diamonds. What is the cause of this?

Reply 13: Coulomb diamonds of devices **A**, **B** and **D** are in fact closing. The low-bias current in device **C** is strongly suppressed as the result of electron-vibrational interactions (it is effectively an example of a Franck-Condon blockade). Importantly, this effect is very well captured by our theoretical model.

This point is now clarified in the revised version of the manuscript (Results and Discussion, page 10):

We further note that, in the case of device **C**, the low-bias current is very strongly suppressed. This is an example of a Franck-Condon blockade,⁴ and is very well captured by our theoretical model.

Reviewer #3 (Remarks to the Author):

Comment 1: The authors study dependence of current through porphyrine molecule noncovalently attached to graphene electrodes on bias voltage, gate voltage, temperature, and anchor group. Experimental data are fitted by theoretical model assuming single electronic level for the bridging molecule, single quantum harmonic vibrational mode coupled to that level, other level-shifting interactions via empirical spectral density, rate theory for electron exchange with electrodes, no other dynamic variables and no other specific interactions with electrodes and environment. The results are interesting and can be published as is. However, judging by novelty and significance, Nature Communications might be inappropriate journal, because I do not see how this paper can influence thinking in the field. The problem is that claims made in the manuscript are not novel but rather elucidate known facts (from reviews and textbooks) using state of the art experimentation and modeling. Let me analyze statements made in the abstract to see what can be a major claim of this work:

Reply 1: We thank the Reviewer for their detailed review and valuable comments, and for stating that "The results are interesting and can be published as is.". We respectfully disagree with their sentiment that our manuscript only "elucidate[s] known facts" and is unlikely "influence thinking in the field". As highlighted in the revised version of the manuscript and emphasised in the reply to all three Reviewers, our work combines a novel theoretical approach with unprecedented experimental access and control to rigorously and unambiguously show that the state-of-the-art of modelling single molecule transport experiments is generally insufficient, and fails for the class of devices which we here consider. We hope this will convince the Reviewer that our carefully and thoroughly conducted study represents a major and significant advance that justifies publication in Nature Communications.

Comment 2:

1) "We observe a simultaneous breakdown of quantum coherent Landauer and semi-classical Marcus theory."

The bridging molecule is large and flexible and the contacts are weak, so that no ballistic transport is expected according, e.g., to p.20 of Ref.16. At the same time semi-classical Marcus theory is always broken-down at low enough temperature according to Section 6.4.3 of the book [V May, O Kuhn, Charge and energy transfer dynamics in molecular systems (Wiley, 2004)], and for pi-conjugated molecules this critical temperature is lower than their thermal decomposition temperature.

Reply 2: *The Reviewer is indeed correct. Nonetheless, the classical Marcus theory and the non-interacting Landauer approach are the two approaches most commonly used to model charge transport through molecular junctions. They are often used in a rather uncritical fashion [see for instance Yuan et al. Nat. Nanotechnol. 13, 322–329 (2018) or Chen et al. Nat. Nanotechnol. 12, 797–803 (2017)]. Therefore, while (from a theoretical perspective) the above statement is hardly surprising, we believe that an experimental verification of this claim, and an effective way to resolve the issue, constitute an important contribution to the field.*

Comment 3:

2) "We propose a theoretical model based on a quantum master equation,..."

All ingredients of the proposed model are known (they are properly cited in the current work).

Reply 3: *This statement has been re-worded in the revised version of the manuscript (Abstract) to state that we use a quantum master equation model, as opposed to propose a quantum master equation model.*

Comment 4:

3) "... and demonstrate that it quantitatively describes rates of electron transfer in single molecules."

Because the model is empirical (fitted to experiment), it is designed to be quantitatively good.

Reply 4: *The theoretical model used here was derived for an Anderson-Holstein-type system, it is therefore not empirical in the sense that its origins are not based purely on empirical observations.*

Furthermore, while the model is indeed fitted to the experiment that does not necessarily imply it will quantitatively describe the observed behaviour. In fact, we have also fitted our experimental data to the classical Marcus (Figure 4, plus many SI, such as Figures S18, S19), single-mode Franck-Condon model (Figure 2) and Landauer models (Figure 2). None of them resulted in a quantitatively (or qualitatively) good fit, as discussed in the manuscript and SI. Our fitting parameters – the tunnel coupling to source and drain and the outer-sphere reorganisation energy – are the only parameters that cannot be derived from first principle due to the random orientation of the molecule in the junction. All other values and functional forms in our model are derived from first principle.

Comment 5:

4) "We show that nuclear tunnelling enhances the rates of low-energy electron transfer,..."

This the consequence of use of the correct phonon correlator in Eq.5 instead of a priori incorrect high-temperature replacement of $\coth(x)$ by $1/x$.

Reply 5: *The Reviewer is correct in pointing out that the shortcomings of the classical treatment stem from the (invalid) approximation in the phononic correlation function. This approximation is however widely used in the literature (whenever the Marcus description of transport is used) without justification.*

Comment 6:

5) "... and demonstrate that the rates are sensitive to both the outer and inner-sphere environmental interactions."

This is the most common situation already recognized in the review [R A Marcus, Chemical and electrochemical electron-transfer theory, *Annu Rev Phys Chem* 15, 155 (1964)].

Reply 6: *While the importance of both the outer and inner sphere contributions has been long recognised in the field of electron transfer, the outer sphere background is typically ignored in experimental studies of charge transport through molecular junctions (as also correctly pointed out by Reviewer 2). See for instance: C. S. Lau et al. Nano Lett. 16 (1), 170-176 (2016); E. Burzurí et al. Nano Lett. 14 (6), 3191-3196 (2014); S. Ballmann Phys. Rev. Lett. 109 (5), 056801 (2012).*

Comment 7:

6) "We find that the nuclear dynamics accompanying electron transfer must be treated quantum mechanically as the quantitative validity of Marcus theory is expected to occur at temperatures exceeding 298K."

See the above discussions.

Reply 7: *It is correct that from a theoretical viewpoint such a statement is hardly surprising. However, the fact that the conventional Marcus theory constitutes a poor theoretical model of molecular conduction needs to be demonstrated to the molecular-electronics community. Theoretically, we have the freedom to choose the most complex models, but experimentally we must typically resort to simplifying assumptions and approximations. In our work, we show that, due to its underlying assumptions, Marcus theory constitutes a poor theoretical description of the experimental transport behaviour.*

Conversely, we provide experimental evidence of nuclear tunnelling in single-molecule junctions. We have further experimentally demonstrated that the non-trivial conductance properties of such a system can only be explained by an interplay between the outer- and inner-sphere vibrational coupling (while accounting for nuclear tunnelling) and electron-electron interactions. We are therefore convinced that the theoretical description validated here should be broadly applicable throughout the field of molecular electronics.

Comment 8: To summarize, all the statements made by the authors are correct and well supported in this work but not novel enough to influence thinking in the field.

Reply 8: *We hope to have convinced the reviewer that even though the individual phenomena observed in our work have been previously discussed in the literature, understanding their simultaneous effect on the transport properties of molecular junctions (which in itself is a non-trivial theoretical problem) constitutes an important advance in the field.*

Comment 9: There is also one comment concerning DFT calculations: low-frequency modes in organic molecular systems are usually substantially anharmonic (with nonlinear vibronic coupling) and intermix with other low-frequency intra- and intermolecular modes. For this reason the assignment (on p.3) of 4.9 meV vibronic progression with 6.0 meV mode calculated for a molecular fragment at some point of its potential energy surface sounds statistically meaningless.

Reply 9: *We thank the Reviewer for pointing this out. This issue is now briefly discussed in the revised version of the manuscript by modifying the following text (Results and Discussion, page 3).*

From:

This interpretation is substantiated by the fact that the average spacing between the lines measured for device **A** corresponds to 4.9 ± 0.3 meV, which is in close agreement with the only strongly-coupled low-lying vibrational mode of the molecule predicted by DFT calculations (6.0 meV, see SI).

To:

This interpretation is supported by the fact that the average spacing between the lines measured for device **A** corresponds to 4.9 ± 0.3 meV, which is in a rough agreement with DFT calculations which predict the presence of a strongly-coupled low-energy vibrational mode (at 6.0 meV, see SI). We note however that any assignment should be treated with caution due to strong anharmonic effects often observed for low-frequency molecular modes.

Reviewers' comments:

Reviewer #2 (Remarks to the Author):

Although authors have made the paper clearer, I still feel that they try to sell their work as if their approach is completely new and that they provide the final answer to the problem. The title is phrased like that but also in the main text additional examples can be found:

- 1) Line 49: the authors write "fully elucidate": the paper does not do that in my opinion as it does not incorporate all the different electrode materials and substrates.
- 2) The references to previous work does not seem to be complete; see e.g. Nature Communications 4, 1710 (2013) and references therein.
- 3) Page 3: Why is the N-1 to N-2 transition the most logical one? (on page 8, the N/N-1 transition is considered to be the most likely one).
- 4) Line 69: I do not agree with the observation of the authors that the lines run parallel to the edges. They clearly do not. It may be an interesting feature and the authors should just present their data in an objective way.
- 5) Line 147: the authors claim excellent agreement with the data but give little insight in the assumption made or to what extent these would be sensitive for the outcome of the calculations. For example, how crucial is the shape of the background spectral density? Would it be the same for all samples? Is there a difference to be expected between gold and graphene leads in this respect?
- 6) Line 234: the authors state that precisely control the environment. It is well known that with their fabrication technique this cannot be done.
- 7) I really don't understand what the authors want to say with the last two lines of the manuscript. Why is it a novel tool and what is meant by extended molecular systems?

Overall, I am in favour of publication of the paper but the above mentioned points should be addressed first.

Reviewer #3 (Remarks to the Author):

In the detailed response of the authors I could not find an explanation how this work can "influence thinking in the field". For this reason I continue to think that Nature Communications might be inappropriate journal.

Reviewer #4 (Remarks to the Author):

The authors present and analyse results of charge transport through graphene-based Zn porphyrin junctions. The main theme is that resonant transport in this weakly-coupled limit realised experimentally can only be fitted accurately by a model which includes several interactions in order to describe non-adiabatic sequential electron transfer events using a QME.

In general, reviewers favourably highlight the importance of the topic and the results presented, however there are questions about the novelty of the theoretical analysis and its significance for the broader scientific community.

My opinion is closer to reviewers 2 and 3. The paper shows that a better fit is obtained when including outer-sphere reorganisation (the most important new element in the model) but I do not see how this goes beyond known theories and models in a significant way to influence the broader audience of

Nature Communications. In the paper and reply to reviewers, the authors draw attention to the need to include electron-electron, electron-vibration and reorganisation energy (inner and outer-sphere) but the methods to do this are all known already. The abstract and reply mention an “interplay” between these interactions but I found this too vague to clearly show that it advances the big picture.

In summary, I believe the paper makes a clear case by showing how some of the models traditionally used fail to properly describe experimental data, however I am not convinced its significance is sufficient to warrant publication in Nature Communications.

Reviewer #5 (Remarks to the Author):

The authors present a study of resonant charge transport through single-molecule junctions that has some very good points, but cannot be published in its current form, in my opinion.

The authors claim a full-quantum new theoretical model to describe weakly-coupled molecular junctions, but the claim does not correspond to the facts, as they are presented. It is true that nondiagonal terms can be eliminated from the quantum master equation (in the absence of accidental degeneracies of a certain kind) by incorporating their effects into the expressions of the rates connecting the diagonal elements. However, here the authors are considering a weakly-coupled molecular junction, in which the effective molecular electronic level involved in the rates can be identified, or at least well approximated, with a physical molecular level. At this point, the expression of the current has nothing substantially different from the (formally identical) one used in ref 11 (eq 11a in ref 11) or previously (for example, eq 20 in *Electrochemistry Communications* 2007, 9, 1343). In fact, the authors use other expressions of the rates into the same current equation.

The authors try to create a gap with previous models to sell better their theoretical setup, but they do some important mistakes in their perspective. For example, the model of ref 11 was proposed to describe systems in conditions and at temperatures for which a classical-type hopping model and the Marcus equation can well be used. Therefore, saying implicitly or explicitly that the model of ref 11 does not apply to a system studied at temperatures for which Marcus theory itself is not (fully) appropriate is at least unfair. This is also misleading, because people who want to use simpler models, where they work very well, could be negatively and mistakenly influenced by this manuscript. Moreover, the same comparison in Fig. S22 shows that a classical-type hopping model with Marcus or improved-Marcus rates also works very well for this system at the still low temperature (compared to what may be needed in devices) of 77 K. This model (or class of models, as resulting from different improvements on Marcus equation) and the model of this study either produce results that, for example, differ on a scale of pAs where the current is on a scale of nAs, or both models (or classes of models) fail appreciably. Fig. S22 shows that using the much more complicated model proposed by part of the same authors in 2018 and used in this study is not worth the effort, indeed, if the aim is to reproduce the I-V results shown in this manuscript for temperatures such as 77 K, which are still lower than may be useful in most practical applications.

What is actually, substantially done in a large part of this study (given the weak coupling to the electrodes and the actual hopping mechanism) is a comparison between Marcus equation and other formulations that improve on it at lower temperatures. This study could be a very good one (once vastly and suitably rephrased in the presentation) as a work enabling exploration of the limits of Marcus equation within a (well-known) kinetic model for the study of molecular junctions in the molecule-electrode weak coupling limit (in addition to the results on page 3). In this perspective, the theoretical fit to the experiments would extract this information from the experiments and show it in a nice and clear way. (Yet, another system should be used by the authors in a next study to show an appreciable failure of the Marcus theory).

Therefore, in my opinion for this study to be accepted for publication, its presentation needs to be

widely reconsidered. Some suggestions are below. Moreover, certain theoretical statements need to be made more precise (please see below).

Page 1, abstract

A model works if it can describe the system or class of systems and physical conditions for which the model was conceived. Therefore asserting that a model fails in general just to support another model is not fair. This is the spirit of some statements in the abstract and elsewhere, and should disappear in my opinion.

Per se, it is well-known that "resonant charge transport through weakly-coupled molecular junctions can be understood as a sequence of non-adiabatic electron transfer steps". So, the pertinent statement in the abstract needs to be somewhat rephrased. In addition, this sentence may be misleading if the kind of resonant charge transport is not better specified. For example, if it is resonant tunneling, there is no such a thing as sequential steps, as the authors well know, since virtual steps are then involved and cannot be separated (with respect to the limits imposed by the Heisenberg uncertainty principle).

Pages 2-4

The statement: " Yet, they do not account for the quantum mechanical ... has long been understood)" regarding refs. 11 and 12 is wrong. Ref 11 proposes a model of charge transport in which the molecule-electrode couplings are weak enough that full reorganization of the molecular system can occur between sequential electron transfer steps. This is also the case in this study, in the facts. Also, I see confusion between redox properties and charge transport. The redox property of molecular junctions is not considered in ref 11, but in JACS 2013, 135, 9420 (same authors), where two charge transport channels are considered: the changing in redox state, which is involved in one of the two channels, contributes negligibly to the conduction but influences the transport through the other effective channel through electron-electron interaction. The authors neglected this study.

Related to the above is the fact eq 1 also easily arises from a classical-type master equation, as it results from the quantum master equation in conditions that lead to only nonzero diagonal terms and describe sequential hopping. And the results of Fig. S22 support, indeed, the fact that an actual classical master equation can be used for the system under study.

Nuclear tunneling is unimportant in the systems/conditions that can be studied using the models of ref 11, but the authors stress the importance of nuclear tunneling in other systems at this point of the manuscript to support the significance of their study.

Page 4, Eq 1

The authors should shortly discuss the meaning of this equation in their theoretical setup, since they insert different rates into the same equation and this equation represents an effective molecular electronic level coupled to the electrodes by a classical-type master equation. If the authors consider that their effective rates make the equation quantum, then they should explain how they compare with the same equation using Marcus-type rate expressions.

Page 4, Fig. 1

The fact that the lines run parallel to the edges should be softened: it is so approximately. This could also justify the non perfect pertinent theoretical fit.

Page 5

Considering the Landauer model or sequential steps is not only a matter of off-resonance, as the authors seem to believe. The sequential mechanism can also be at play off-resonance. In fact, depending on the system, changing the bias or gate voltage can bring the molecular bridge into conditions of resonance, and the same kinetic equations and rates can be applied to describe the

system passing from off-resonance to resonance conditions.

Current asymmetry has also been obtained in other models; for example: Phys. Rev. Lett. 1997, 79, 2530 and the same ref 11. The authors should acknowledge this.

Pages 8-12 and Fig. S22

The authors compare with Marcus-Jortner and low-temperature corrected Marcus theory in the SI, where the better performance of their method compared to the latter is declared but not clear at all from Fig. S22. Also, declaring the better performance of a method over another for graphs where, for example, I is in nA and ΔI is in pA requires much more caution at least, all the more considering that the authors are using continuous models to describe sets of discrete experimental points collected during voltage ramps.

Reply to Reviewers' Comments and Details of Revisions

Reviewer #2 (Remarks to the Author):

Comment 1:

Although authors have made the paper clearer, I still feel that that try to sell their work as if their approach is completely new and that they provide the final answer to the problem. The title is phrased like that but also in the main text additional examples can be found:

Reply 1: *We are glad Reviewer #2 found the paper clearer following our revisions. We have addressed their remaining recommendations below, and slightly modified the title. We hope that they will now support the publication of our manuscript in Nature Communications.*

Comment 2:

1) Line 49: the authors write “fully elucidate”: the paper does not do that in my opinion as it does not incorporate all the different electrode materials and substrates.

Reply 2: *The Reviewer is correct here that the statement in question may be too strong. For example, if the coupling of the molecular system to the electrodes is very strong then the used (non-adiabatic) model is no longer valid. Therefore we have changed the title to reflect that our approach is for weakly coupled single-molecule junctions. We do believe however our model captures all the components necessary to describe resonant electron transfer in weakly coupled single-molecule junctions with different substrates and electrode materials (in the absence of molecular excited states). However, we agree that until this has been verified experimentally we should qualify our statements so that it refers specifically to our device set-up. We have removed the term “fully elucidate” from the Introduction.*

See also our response to Comment 6 for additional text added on the subject of different substrates.

Comment 3:

2) The references to previous work does not seem to be complete; see e.g. Nature Communications 4, 1710 (2013) and references therein

Reply 3: *We thank the reviewer for this suggestion and have added the following references to the introduction.*

- Nat. Comm. **4**, 1710 (2013)
- Phys. Rev. B **79**, 115203 (2009)

Comment 4:

3) Page 3: Why is the N-1 to N-2 transition the most logical one? (on page 8, the N/N-1 transition is considered to be the most likely one).

Reply 4: *Since graphene on HfO₂ (and indeed on SiO₂) is p-doped and the molecular anchor groups are electron-rich, we assign the closest transition to the Fermi level of the leads ($V_g = 0$) to be the N-*

1/N transition where the N-charge state corresponds to the neutral molecular species. For devices B–D we study the closest transition to $V_g=0$, and therefore identify it as N-1/N. For Device A (Fig 1/Fig 2), the transition we study in detail is the second closest to $V_g=0$, as shown in the stability diagram in the SI, and therefore it is identified as N-2/N-1. Unfortunately, the device A became too unstable after the temperature-dependent measurements to study the N-1/N transition in detail.

Furthermore, we have developed a method to assign charge states of single-molecule transistors based on the magnitude of the current in the resonant transport regions. The method is detailed in a publication that is currently under peer-review, we have attached the manuscript for the Reviewers to read. In brief, asymmetries in the current (due to spin degeneracy and asymmetric molecule/lead coupling) result in the highest current of the resonant tunnelling region to reside on the side of the Coulomb diamond corresponding to a doublet (N-1) state. This new work confirms the assignment we have given in this paper, and allows us to be confident as to which charge states are involved.

Comment 5:

4) Line 69: I do not agree with the observation of the authors that the lines run parallel to the edges. They clearly do not. It may be an interesting feature and the authors should just present their data in an objective way.

Reply 5: *To obtain the line graph in Fig 1c we average the data in the high conductance region along a series of lines that run parallel to the edge of the Coulomb diamond and plot these as a function of potential. The peaks we observe in Fig 1c would not be present if the lines did not run, at least approximately, parallel. Nonetheless, we have changed the wording in the following sentences (Results, page 4/5) regarding whether the lines are parallel. In addition, we believe that the origin of these lines is vibrational because: 1) fluctuations in the DOS do not give stepwise increases in the current, but rather regions of increased conductance alternated by regions of negative differential conductance, which we do not observe, and 2) the spacing between the lines is approximately equal, which is a feature of molecular modes and overtones, and unlikely to be present in DOS fluctuations.*

From:

We further observe lines of increased conductance running parallel to the edges of the high-current region.²⁰ Previous studies^{6,7,21} have assigned such conductance lines to vibrational excitations of the molecule during the charging process.

We note that density of states fluctuations in the leads cannot explain the observed behaviour, because they would give rise to lines non-parallel to the edges of the Coulomb diamond.²⁰ The features observed here are on the other hand parallel to the edges of the diamond (as is more clearly visible in Figure S15 in the SI).

To:

Inside the sequential tunnelling region, we observe lines of increased conductance (Figure 1c), which are spaced equally apart. We are able to assign these conductance lines to vibrational excitations of the molecule during the charging process, in line with previous studies^{12,13,23}.

The assignment of the conductance lines to molecular vibrations, as opposed to e.g. density of states (DOS) fluctuations in the graphene²⁴, is robust despite the presence of some imperfections in the experimental data (such as jumps in the edges of the Coulomb

diamond) for several reasons. Firstly, the line graph in Figure 1c is the data in the high conductance region averaged along a series of lines that run parallel to the edge of the Coulomb diamond and plotted as a function of potential: the peaks we observe would not be present if the lines did not run, at least approximately, parallel to the edges of the high conductance region. Furthermore, fluctuations in the DOS do not give stepwise increases in the current, but rather regions of increased conductance alternated by regions of negative differential conductance, which we do not observe. The spacing between the lines is approximately equal, which is a feature of molecular modes and overtones, and unlikely to be present in DOS fluctuations. Finally, we found the same equally spaced conductance lines in another device (device E, see SI).

Comment 6

Line 147: the authors claim excellent agreement with the data but give little insight in the assumption made or to what extent these would be sensitive for the outcome of the calculations. For example, how crucial is the shape of the background spectral density? Would it be the same for all samples? Is there a difference to be expected between gold and graphene leads in this respect?

Reply 6: We have chosen to model the outer-sphere (background) coupling using a super-Ohmic spectral density with an exponential cut-off (as discussed in the SI):

$$J(\omega) = \frac{\lambda_o}{2} \left(\frac{\omega}{\omega_c} \right)^3 e^{-\omega/\omega_c}$$

As we now discuss in the revised version of the manuscript, such a spectral density describes deformation coupling of localised charges to the bulk phonons, and is commonly used in the literature (see discussion in Ref. 27 and 28).

This background spectral density is parameterised by the cut-off frequency ω_c and the outer-sphere reorganisation energy, λ_o . We find empirically that the quality of the fitting is indeed sensitive to the choice of cut-off frequency of the background spectral density, this is addressed in Figure S14. A choice of $\hbar\omega_c = 25$ meV gives the best quantitative agreement to experiment and is used to model devices **B–D** in the main text and **E–M** in the SI. Therefore this seems to be an intrinsic property of the SiO₂ substrate and for all devices made with this substrate the value of cut-off frequency would be approximately the same. These points are addressed in Section S5.

For different substrates, or environments such as liquid solvent, we expect a different value of the cut-off frequency to be the best choice, perhaps even a modified function of $J(\omega)$. We acknowledge that this will need to be verified experimentally by future studies.

The reorganisation energy, λ_o , (which quantifies the molecule-substrate vibrational coupling strength) will depend on the orientation and position of the molecule on the substrate and should be expected to vary between different devices. Therefore, we have used λ_o as a free fitting parameter.

We assume the wide-band approximation throughout (i.e. a constant density of states in the leads which appears to be valid within the considered bias window for strongly-doped graphene electrodes) and therefore we do not expect any differences between the modelling procedures whether graphene or gold electrodes are used (although we note that the molecule-lead coupling can be significantly stronger in the case of gold electrodes). There may be differences in the contributions

of the gold and graphene in terms of their ability to screen charges etc. but this will be incorporated into the modelling of the outer-sphere contributions to the electron transfer rates.

We have modified the following text (Results, High-bias studies, page 12) to reflect these points more clearly.

From:

The current–voltage traces of devices **B–D** in Figure 4a are therefore fitted using the QME approach and with spectral density given in equation 7, taking only Γ_S , Γ_D and λ_o as free fitting parameters (see SI section 5 and 8).

To:

The current–voltage traces of devices **B–D** in Figure 4c are therefore fitted using the quantum approach and with spectral density given in equation 7, taking only Γ_S , Γ_D and λ_o as free fitting parameters. The cut-off phonon energy, $\hbar\omega_c$, is fixed at 25 meV, we expect this parameter to be intrinsic to the SiO₂ substrate, (see SI section 5 and 8 for the dependence of the fitting on $\hbar\omega_c$).

We have also changed the following text in the Conclusions (page 15):

From:

We believe that the theoretical description validated here should be broadly applicable throughout the field of molecular electronics.

To:

We have shown that all the ingredients of our quantum model are necessary to develop a quantitative description of resonant transport through weakly coupled single-molecule junctions, especially at low temperature. Therefore, we believe that the theoretical description validated here should be broadly applicable throughout the field of molecular electronics.

Comment 7:

6) Line 234: the authors state that precisely control the environment. It is well know that with their fabrication technique this cannot be done.

Reply 7: *We did not mean to claim that we were controlling the environment precisely using our protocol for device fabrication. One of the main results of our work was that: a) outer-sphere reorganisation contributes significantly to the charge-transport properties of our devices and b) it varies from device-to-device (we obtained outer sphere reorganisation energies from 110 – 250 meV). We meant to state that if one wants to fabricate single-molecule devices with more reproducible charge-transport properties, then one must strive to control the environment surrounding the molecular structure. We feel this is an important avenue for future research on single-molecule devices both from a fundamental and reproducibility standpoint. We have altered the text (Conclusions, page 15) to clarify this, and make a suggestion as to how the control could be achieved with two additional references.*

From:

Our results further demonstrate the importance of precisely controlling the (too often ignored) molecular outer-sphere environment when designing functional molecular technologies such as molecular transistors, diodes and thermoelectric materials.

To:

Our results further demonstrate that in the design of functional molecular technologies such as molecular transistors, diodes and thermoelectric materials, attempts must be made to precisely control the (often ignored) molecular outer-sphere environment. This could be achieved by, for example, synthesizing supramolecular assemblies that isolate the molecular structure from the local environment^{35,36}.

Comment 8:

7) I really don't understand what the authors want to say with the last two lines of the manuscript. Why is it a novel tool and what is meant by extended molecular systems?

Reply 8: *Traditionally, in order to study electron transfer either a large number of molecules adsorbed at the solution/electrode interface are studied in an electrochemical set-up, or photo-induced electron transfer experiments are carried out on dilute molecular solutions. The use of a three-terminal single-molecule junction allows us to isolate a single-molecule, tune a charge transition into resonance, and then investigate the influence of electronic coupling, electron-electron interactions and relative contributions of inner/outer sphere reorganisation at different temperatures. This is what we meant by 'we have also shown that single-molecule junctions can act as a novel tool to unravel the mechanism of individual electron transfers in molecular systems'. We have removed the word novel to avoid any confusion that we believe single-molecule junctions are novel. Furthermore we have revised the final sentence (page 15) to reflect a valuable and more tangible future direction.*

From:

Finally, we have also shown that single-molecule junctions can act as a novel tool to unravel the mechanism of individual electron transfers in molecular systems. This opens the door towards investigations of mechanisms of electron transport within extended molecular systems as well as photo-induced electron transfer, and thus shedding further light on some of the most important phenomena in chemistry.

To:

Finally, we have also shown that single-molecule junctions can act as a tool to unravel the mechanism of individual electron transfers in molecular systems. This opens the door towards precise single-molecule experimental investigations of the influence of various liquid, solid or supramolecular environments on the rates of heterogeneous electron transfers. A comprehensive understanding of the influence of the local environment on electron transport could have significant impact on improving reproducibility in single-molecule electronics or optimising the performance of thin-film organic electronic devices.

Overall, I am in favour of publication of the paper but the above mentioned points should be addressed first.

We are glad that the Reviewer now supports the publication of our manuscript in Nature Communications and hope that the revisions discussed above fully address their remaining concerns.

Reviewer #3 (Remarks to the Author):

Comment:

In the detailed response of the authors I could not find an explanation how this work can "influence thinking in the field". For this reason I continue to think that Nature Communications might be inappropriate journal.

Reply: *In our work, we have demonstrated that the theoretical approaches commonly used to model charge transport through molecular junctions are in practice insufficient, especially at low-temperature conditions considered in our experimental studies. Instead, we have shown that in order to accurately describe resonant transport through weakly-coupled molecular junctions at low temperatures it is necessary to **simultaneously** account for:*

- *inner-sphere vibrational interactions,*
- *outer-sphere environmental coupling,*
- *lifetime broadening,*
- *the effects of electron-electron interactions.*

To the best of our knowledge, all existing experimental studies in this area ignore at least one of the above effects. This may be why typically only at most qualitative agreement between the experimental data and theoretical modelling is obtained in these systems, see for instance:

- *E. Burzurí et al. ACS Nano 10.2 (2016): 2521-2527.*
- *C. S. Lau et al. Nano Lett. 16.1 (2015): 170-176.*
- *E. Burzurí et al. Nano Lett. 14.6 (2014): 3191-3196.*
- *H. Park et al. Nature 407.6800 (2000): 57*
- *S. Braig and K. Flensberg Phys. Rev. B 68.20 (2003): 205324*

Furthermore, our research empirically demonstrates the influence of the wider molecular environment on the charge transport properties on a single-molecule junction, and we could not find examples in the literature where attempts have been made by experimentalists to control this potentially crucial contribution. A major challenge in single-molecule electronics is device-to-device reproducibility and one of our conclusions is that controlling the molecular environment should be an avenue of future research for the community. For these reasons, we emphatically disagree with the Reviewer's assessment and strongly believe that this work will "influence thinking in the field".

Reviewer #4 (Remarks to the Author):

Comment:

The authors present and analyse results of charge transport through graphene-based Zn porphyrin junctions. The main theme is that resonant transport in this weakly-coupled limit realised experimentally can only be fitted accurately by a model which includes several interactions in order to describe non-adiabatic sequential electron transfer events using a QME.

In general, reviewers favourably highlight the importance of the topic and the results presented, however there are questions about the novelty of the theoretical analysis and its significance for the broader scientific community.

My opinion is closer to reviewers 2 and 3. The paper shows that a better fit is obtained when including outer-sphere reorganisation (the most important new element in the model) but I do not see how this goes beyond known theories and models in a significant way to influence the broader audience of Nature Communications. In the paper and reply to reviewers, the authors draw attention to the need to include electron-electron, electron-vibration and reorganisation energy (inner and outer-sphere) but the methods to do this are all known already.

The abstract and reply mention an “interplay” between these interactions but I found this too vague to clearly show that it advances the big picture.

In summary, I believe the paper makes a clear case by showing how some of the models traditionally used fail to properly describe experimental data, however I am not convinced its significance is sufficient to warrant publication in Nature Communications.

Reply: *While the Reviewer is correct in asserting that methods to incorporate the aforementioned interactions into the models for charge transport have been known, we believe that the experimental demonstration of theoretically known phenomena has intrinsic scientific value, there are many examples of this (e.g.: Nature Nanotechnology 13, 376–380, (2018)). As described in the Reply to Reviewer 3 we could not find any combined theoretical or experimental studies that incorporate the influence of inner-sphere vibrational interactions, outer-sphere environmental coupling, lifetime broadening, and the effects of electron-electron interactions. We have been able to do this on the single-molecule level and unpick the interactions step-by-step, see Fig. 2a, Fig. S16 and Results page 6 and 7 respectively. We have demonstrated that they are all required if one wishes to quantitatively describe sequential electron transport in weakly coupled molecular junctions in a quantitative way especially at low temperature, this has not been done before.*

Furthermore, upon the suggestion of Reviewer 5, we have been able to investigate the temperature correspondence of our quantum model with a classical Marcus theory model for a single-molecule junction and compared to experimental data.

Overall, we believe the paper demonstrates one of the most complete combined theoretical/experimental studies of sequential electron transport in weakly coupled molecular junctions to date. Our approach can be extended to new substrates, environments and molecular structures. Therefore it is an important contribution to the fundamental study of electron transfer and transport. We hope that considering the re-framing of the theoretical description and results, and the new analysis of the temperature dependence, that the Reviewer will now support the publication of this manuscript in Nature Communications.

Reviewer #5 (Remarks to the Author):

Comment 1:

The authors present a study of resonant charge transport through single-molecule junctions that has some very good points, but cannot be published in its current form, in my opinion.

Reply 1: *We thank the Reviewer for their very insightful comments which have led to significant improvements to the manuscript. In particular, in the light of their comments, parts of this manuscript were restructured and the focus of the manuscript has shifted to accommodate the Reviewer's suggestions. We hope that the Reviewer will find this revised version of the manuscript suitable for publication in Nature Communications.*

Comment 2:

The authors claim a full-quantum new theoretical model to describe weakly-coupled molecular junctions, but the claim does not correspond to the facts, as they are presented. It is true that nondiagonal terms can be eliminated from the quantum master equation (in the absence of accidental degeneracies of a certain kind) by incorporating their effects into the expressions of the rates connecting the diagonal elements. However, here the authors are considering a weakly-coupled molecular junction, in which the effective molecular electronic level involved in the rates can be identified, or at least well approximated, with a physical molecular level. At this point, the expression of the current has nothing substantially different from the (formally identical) one used in ref 11 (eq 11a in ref 11) or previously (for example, eq 20 in *Electrochemistry Communications* 2007, 9, 1343). In fact, the authors use other expressions of the rates into the same current equation.

Reply 2: *The expression for the electric current used in our study was derived using a generalised quantum master equation in Ref. 22 (now Ref. 5 in the revised manuscript). As the Reviewer correctly points out, in the present case (of a single, spatially non-degenerate electronic level), the expression for the electric current obtained in this way (Eq. 1) has effectively a well-known semi-classical (rate-equation) structure.*

As the Reviewer also correctly points out, the difference between the method used here and earlier studies (cited above by the Reviewer) lies in the form of the electron transfer rates and molecular densities of states (DOS), $k_{red/ox}$, given in (Eq. 2, 3, 4). Unlike in the case of, for instance, Ref. 11 (now Ref. 8 in the revised manuscript), the molecular DOS in (Eq. 4) account for the structure of the vibrational spectral density, nuclear tunnelling effects and lifetime broadening. We agree that, while it is important to highlight the novelty of this work, it is also important to clarify the connection to the earlier studies present in the literature.

The introduction of (Eq. 1) (now in the Introduction, page 2) has therefore been changed with the references suggested by the Reviewer now included:

From:

The general expression (quantum master equation, QME) for the net current is given

by:^{3,10,24}

To:

The expression for the net current through a weakly coupled molecular junction has a well-known form⁴⁻⁸.

We have altered the Introduction (pages 2–4) to make it clear that it is the quantum (Eq. 4) or classical (Eq. 6) formulations of $k_{red/ox}$ that we test against our experimental data.

Furthermore, we have removed all 18 instances of references to the (quantum-master equation or QME) from text, captions and figures to avoid confusion and changed the wording simply to ‘quantum model’ or ‘quantum’ if we are referring to the use of a quantum formulation of $k_{red/ox}$. One such example (Results, High-bias studies, page 11) is given below:

From:

These outer and inner-sphere contributions are plotted in Figure 3b. Figure 3c shows the comparison between the QME and MT reduction rates calculated for the instructive values of λ .

To:

These outer and inner-sphere contributions are plotted in Figure 4a. Figure 4b shows the comparison between the quantum and MT molecular DOS calculated for the instructive values of λ .

Comment 3:

The authors try to create a gap with previous models to sell better their theoretical setup, but they do some important mistakes in their perspective. For example, the model of ref 11 was proposed to describe systems in conditions and at temperatures for which a classical-type hopping model and the Marcus equation can well be used. Therefore, saying implicitly or explicitly that the model of ref 11 does not apply to a system studied at temperatures for which Marcus theory itself is not (fully) appropriate is at least unfair. This is also misleading, because people who want to use simpler models, where they work very well, could be negatively and mistakenly influenced by this manuscript. Moreover, the same comparison in Fig. S22 shows that a classical-type hopping model with Marcus or improved-Marcus rates also works very well for this system at the still low temperature (compared to what may be needed in devices) of 77 K. This model (or class of models, as resulting from different improvements on Marcus equation) and the model of this study either produce results that, for example, differ on a scale of pAs where the current is on a scale of nAs, or both models (or classes of models) fail appreciably. Fig. S22 shows that using the much more complicated model proposed by part of the same authors in 2018 and used in this study is not worth the effort, indeed, if the aim is to reproduce the I-V results shown in this manuscript for temperatures such as 77 K, which are still lower than may be useful in most practical applications.

Reply 3: We agree that our earlier discussion of the semi-classical approaches (for instance that of Ref. 11 (now Ref. 8)) might have been somewhat misleading. As the Reviewer correctly points out, Marcus-type approaches constitute an attractive theoretical framework for describing charge transport through molecular junctions at higher temperatures. The (partial) failure of the Marcus-type theories in our case stems from the low-temperature conditions used in our experimental study.

Indeed, as we clearly state in the revised version of the manuscript, one should expect Marcus theory to correctly capture the charge transport mechanism at temperatures higher than used in our study. Furthermore, as the Reviewer correctly points out, the improved versions of Marcus theory can

correctly describe transport properties of some of the devices at 77 K. However, as discussed in more detail in Reply 12 below, the quantum model gives the most robust description of the data for devices B–M at 77 K.

We note that the low-temperature (3 or 5 K) measurements are clearly beyond the scope of such semi-classical (Marcus-type) theories. Our theory (from Ref.5) while more complicated than the high-temperature alternatives, is capable of describing charge transport across the entire relevant temperature range.

The manuscript has been revised to address the above points. The introduction of MT (Introduction, pages 2–4) now explicitly states that MT is: typically applicable at ambient conditions (as confirmed experimentally^{10,11}) this approach is expected to break down at cryogenic temperatures where lifetime broadening and the quantum nature of the vibrational motion become relevant.

In addition (Results, Temperature-dependence, page 11) we have state that: we expect that in general MT is an adequate model for electron transfer in weakly coupled molecular systems at 298 K.

Furthermore we have altered the following sections of text (Results, High-bias studies, page 12) to more fairly discuss MT:

From:

We find that our experimental charge transport data are not adequately described by MT, as shown in Figure 4 for devices **B–D**.¹⁹

To:

We find that our experimental charge transport data for devices **B–D** can be described by MT with appreciable errors in the fits compared to the magnitude of the current, as shown in Figure 4c²², and there are features of the charge transport that are not captured by this approach.

From (Results, High-bias studies, page 13):

The results (presented in the SI) show the transport behaviour of these devices can also be successfully explained using our QME model (and not using the semi-classical Marcus theory).

To:

The results (presented in the SI) show the transport behaviour of these devices can also be successfully explained using our quantum model.

From (Conclusions, page 15):

We have further shown that neither the conventional Landauer and Marcus theories nor the single-mode Franck-Condon model provide an accurate theoretical description of the experimentally-observed charge transport.

To:

In contrast, neither the conventional Landauer theory nor the single-mode Franck-Condon model provides an accurate theoretical description of the experimentally-observed charge transport. We have further shown that at cryogenic temperatures (below 77 K), Marcus theory also constitutes an inadequate description of the charge transport mechanism due to the importance of nuclear tunnelling under these conditions.

Regarding the discussion of refined versions of MT we have modified the following text (Results, High-bias studies, page 12):

From:

As shown in the SI, such approaches partially rectify some of the shortcomings of MT, highlighting the non-classical mechanism of electron transfer even in these relatively high-temperature conditions. Our experimental data, however, are best described by our fully quantum mechanical treatment involving both inner and outer sphere reorganisation, as discussed above.

To:

As shown in the SI, for some devices such approaches rectify the limitations of MT, (at the expense of additional fitting parameters) and highlight the non-classical mechanism of electron transfer in these relatively high-temperature (for single-molecule devices) conditions. Our experimental data, however, are generally better described by our fully quantum mechanical treatment involving both inner and outer sphere reorganisation, as discussed above.

Comment 4:

What is actually, substantially done in a large part of this study (given the weak coupling to the electrodes and the actual hopping mechanism) is a comparison between Marcus equation and other formulations that improve on it at lower temperatures. This study could be a very good one (once vastly and suitably rephrased in the presentation) as a work enabling exploration of the limits of Marcus equation within a (well-known) kinetic model for the study of molecular junctions in the molecule-electrode weak coupling limit (in addition to the results on page 3). In this perspective, the theoretical fit to the experiments would extract this information from the experiments and show it in a nice and clear way. (Yet, another system should be used by the authors in a next study to show an appreciable failure of the Marcus theory).

Reply 4: *Following the Reviewer's suggestion we have restructured the manuscript to provide more focus on the applicability of Marcus theory in the considered single-molecule junctions. To this end, we have also added a new figure (Figure 3b) which examines the suitability of Marcus-type approaches in the devices considered here. This is done by comparing the zero-bias conductance (on resonance) as predicted by the "Marcus" and "quantum" theories as a function of temperature. The differences in the quantum and Marcus approaches displayed in Figure 3b are discussed in detail in a new subsection (Results, Temperature dependence) and the temperature dependence is rationalised.*

In addition to the new Figure and Subsection, the following text has been added (Conclusions, page 15):

An examination of the temperature dependence of the quantum and Marcus theories suggests that correspondence between the two approaches should be reached in our devices at some point above 100 K, but will depend on the overall value of the

reorganisation energy. We have shown that all the ingredients of our quantum model are necessary to develop a quantitative description of resonant transport through weakly coupled single-molecule junctions, especially at low temperature.

Comment 5: Therefore, in my opinion for this study to be accepted for publication, its presentation needs to be widely reconsidered. Some suggestions are below.

Reply 5: *We have significantly revised our manuscript in accordance with the Reviewer's comments and hope that they will now support the publication of this manuscript in Nature Communications.*

Comment 6:

Moreover, certain theoretical statements need to be made more precise (please see below).

Page 1, abstract

A model works if it can describe the system or class of systems and physical conditions for which the model was conceived. Therefore asserting that a model fails in general just to support another model is not fair. This is the spirit of some statements in the abstract and elsewhere, and should disappear in my opinion.

Reply 6: *Following the Reviewer's suggestions, the statements in question (pertinent to Marcus-type description of charge transport) have been removed or revised in the new version of the manuscript. The details of the revisions are in the answer to Reply 3.*

Comment 7:

Per se, it is well-known that "resonant charge transport through weakly-coupled molecular junctions can be understood as a sequence of non-adiabatic electron transfer steps". So, the pertinent statement in the abstract needs to be somewhat rephrased. In addition, this sentence may be misleading if the kind of resonant charge transport is not better specified. For example, if it is resonant tunneling, there is no such a thing as sequential steps, as the authors well know, since virtual steps are then involved and cannot be separated (with respect to the limits imposed by the Heisenberg uncertainty principle).

Reply 7:

The Reviewer is correct in their comments regarding the statement in question. We have revised it as follows (Abstract):

From:

Instead, we demonstrate that resonant charge transport through weakly-coupled molecular junctions can be understood as a sequence of non-adiabatic electron transfer steps, and describe it using a theoretical model based on a quantum master equation.

To:

Instead, we model the overall charge transport as a sequence of non-adiabatic electron transfers, the rates of which depend on both outer and inner-sphere vibrational interactions.

Comment 8:

Pages 2-4

The statement: “ Yet, they do not account for the quantum mechanical ... has long been understood)” regarding refs. 11 and 12 is wrong. Ref 11 proposes a model of charge transport in which the molecule-electrode couplings are weak enough that full reorganization of the molecular system can occur between sequential electron transfer steps. This is also the case in this study, in the facts. Also, I see confusion between redox properties and charge transport. The redox property of molecular junctions is not considered in ref 11, but in JACS 2013, 135, 9420 (same authors), where two charge transport channels are considered: the changing in redox state, which is involved in one of the two channels, contributes negligibly to the conduction but influences the transport through the other effective channel through electron-electron interaction. The authors neglected this study. Related to the above is the fact eq 1 also easily arises from a classical-type master equation, as it results from the quantum master equation in conditions that lead to only nonzero diagonal terms and describe sequential hopping. And the results of Fig. S22 support, indeed, the fact that an actual classical master equation can be used for the system under study.

Nuclear tunneling is unimportant in the systems/conditions that can be studied using the models of ref 11, but the authors stress the importance of nuclear tunneling in other systems at this point of the manuscript to support the significance of their study.

Reply 8: *We thank the Reviewer for bringing these issues to our attention. Firstly, we concur that nuclear tunnelling can be disregarded in the conditions considered in Ref. 11 (i.e. at relatively high temperature/weak molecular-lead coupling and when the electronic degrees of freedom interact predominantly with the outer-sphere environment). We have re-written our introduction regarding the MT approach outlined in Ref 11 and specific examples of this are given in Reply 3.*

In our case, for our low-temperature measurements (at 3–50 K) considered here, the quantum-mechanical nature of the vibrational degrees of freedom does need to be accounted for in order to correctly model the observed behaviour. Admittedly, at around 77 K the situation is somewhat more delicate. The conventional Marcus theory appears to significantly underestimate the low-bias conductance (for the majority of molecular devices) but the various extensions of Marcus theory (studied in the SI) seem to be able to capture the mechanism of charge transport through most of the considered junctions. As stated before and now also discussed in the manuscript, while one should indeed expect the conventional Marcus description of charge transport to be valid at around room temperature, it clearly does not constitute an accurate description in the low-temperature conditions considered in parts of our work.

We don't feel that we have confused redox properties and charge transport; we found 2 uses of the word 'redox' in the manuscript, both of which were in reference to the porphyrin being a 'redox-active' molecule. We have removed this to avoid any confusion that we believe we are looking at the type of junction that is defined as a redox junction in JACS 2013, 135, 9420. However we have added the reference (JACS 2013, 135, 9420 by Migliore and Nitzan) suggested by the Reviewer and mention redox molecular junctions as part of our introduction to the charge transport we observe in our devices. The following sentence (Introduction, page 2) has been added to the Introduction to reflect the changes:

In contrast to redox molecular junctions³, (in which the charging/discharging of the molecule has no direct contribution to the current) the current that flows through the molecular junction during resonant transport in a weakly coupled junction is a result of these sequential electron transfers to (i.e. a reduction process) and from (i.e. an oxidation process) the molecule.

We use the terms reduction and oxidation extensively to refer to the hopping of an electron on to and off the molecule respectively during sequential transport, which is valid as long as the number of electrons on the molecule (N) remains a good quantum number ($\hbar\Gamma < E_c$).

Comment 9:

Page 4, Eq 1

The authors should shortly discuss the meaning of this equation in their theoretical setup, since they insert different rates into the same equation and this equation represents an effective molecular electronic level coupled to the electrodes by a classical-type master equation. If the authors consider that their effective rates make the equation quantum, then they should explain how they compare with the same equation using Marcus-type rate expressions.

Reply 9: We thank the Reviewer for this suggestion. As they correctly point out, the “quantum” character of our theory stems from the form taken by the electron-transfer rates or, more specifically, the molecular DOS (which account for the quantum character of the vibrational degrees of freedom as well as the lifetime broadening).

The re-written Introduction reframes the theoretical approach to clarify that the quantum and classical nature of the description of current reside in the terms for the molecular DOS. We also make this clear when we introduce the expressions for the molecular DOS, as shown below:

We proceed to quantitatively describe the observed charge transport by accounting for both lifetime broadening and the influence of the vibrational environment in our quantum-mechanical expression for the DOS⁵:

$$k_{red/ox}(\epsilon) = \frac{1}{\pi} \text{Re} \int_0^\infty e^{\sigma i(\epsilon - \mu)t/\hbar} e^{-t/\tau} B(t) dt, \quad (4)$$

And:

At higher temperatures, $k_B T \gg \hbar\langle\omega\rangle, \hbar/\tau$, it is possible to simplify equation 4 by disregarding lifetime broadening and considering a high-temperature limit within the phononic correlation function²⁴. This yields the previously discussed MT in which the molecular DOS takes the familiar classical form^{10,30,31}:

$$k_{red/ox}(\epsilon) = \sqrt{\frac{1}{4\pi\lambda k_B T}} \exp\left[-\frac{(\lambda \pm (\epsilon - \mu))^2}{4\lambda k_B T}\right], \quad (6)$$

Comment 10:

Page 4, Fig. 1

The fact that the lines run parallel to the edges should be softened: it is so approximately. This could also justify the non perfect pertinent theoretical fit.

Reply 10: This statement has been revised accordingly in the new version of the manuscript. The details of our revisions are outlined in our response to Reviewer 2 (Comment & Reply 5).

Comment 11:

Page 5

Considering the Landauer model or sequential steps is not only a matter of off-resonance, as the

authors seem to believe. The sequential mechanism can also be at play off-resonance. In fact, depending on the system, changing the bias or gate voltage can bring the molecular bridge into conditions of resonance, and the same kinetic equations and rates can be applied to describe the system passing from off-resonance to resonance conditions.

Current asymmetry has also been obtained in other models; for example: Phys. Rev. Lett. 1997, 79, 2530 and the same ref 11. The authors should acknowledge this.

Reply 11: *We thank the Reviewer for bringing this to our attention, we agree that the statement in question was somewhat misleading and has been removed.*

There are two sources of asymmetry in our theoretical model: the electron-electron and electron-vibrational interactions (both of which require asymmetric molecule-lead coupling). As discussed in the manuscript, the electron-electron interactions (in the presence of asymmetric molecule-lead coupling) give rise to an asymmetry between the current at positive and negative bias voltages. The electron-vibrational interactions, on the other hand, give rise to asymmetry with respect to the charge degeneracy point [see for instance Figure. 3c in Ref.5 (new manuscript) /22 (old manuscript)].

The Reviewer mentions a different mechanism of current rectification (discussed in the articles mentioned above) which stems from an asymmetric potential distribution in the junction (i.e. asymmetrically applied bias voltage), and which can indeed give rise to significant current rectification also in the off-resonant transport regime.

In our work, the voltage drop across the molecule can be calculated from the gradients of lines enclosing the high conductance region. For device A, $\alpha_S = C_S/C_{tot} = 0.45$, (if $\alpha_S = 0.5$ then bias is applied symmetrically). Therefore we find, but don't assume, that the bias is applied almost symmetrically. Consequently, the observed current asymmetry stems from the asymmetric molecule-lead coupling (in the presence of electron-electron and electron-vibrational interactions) and leads to a factor of 2 in the current at different polarities.

The issues discussed above are clarified in the revised version of the manuscript. Additionally, the previous studies suggested by the Reviewer are cited in this discussion in the revised version of the manuscript (Results, Low Temperature, page 6).

From:

The current-voltage trace of device A measured on resonance (Figure 2a) reveals an asymmetry between the current at positive and negative bias voltages. This is a direct result of electron-electron interactions in the presence of asymmetric molecule-electrode couplings and spin degeneracy. If tunnelling occurs into an unoccupied orbital (LUMO) (e.g. the $N/N+1$ transition, where N is the number of electrons on the molecule in the neutral state) two possible pathways exist for reduction – an electron of either spin can tunnel from the electrode onto the molecule. Only one possible path exists for the oxidation as the unpaired electron (in what is now the SOMO) tunnels out of the molecule and into the electrode. Conversely, if tunnelling occurs into a singly occupied orbital (e.g. the $N-1/N$ transition) the opposite is the case: only electrons of the opposite spin to that on the molecule can reduce the molecule, but electrons of either spin can oxidise the neutral molecule. The number of possible transitions is accounted for by setting Ω to 0 for the $N/N+1$ transition or 1 for the $N-1/N$ transition. The current asymmetry with respect to the sign of the bias voltage cannot be observed in non-interacting (Landauer) systems, i.e. off-resonance or in the case of strong coupling between the molecule and the electrodes

(where the energy uncertainty associated with the lifetime of the electronic states is greater than the energy required to change the charge state of the molecule).

To:

The current-voltage trace of device A measured on resonance (Figure 2a) reveals an asymmetry between the current at positive and negative bias voltages. The potential drop across the molecule is almost symmetric: $\alpha_S = C_S/C_{tot} = 0.45$, where C_S is the capacitance to the source and C_{tot} is the sum of the capacitances to the source, drain and gate. Therefore the current rectification is not due to an asymmetric potential drop across the molecule²⁵. Instead it is a direct result of electron-electron interactions in the presence of asymmetric molecule-electrode couplings and spin degeneracy⁷ (accounted for by Ω), and can be inferred from equation 1–3. The current rectification ratio will be between 1 (for symmetric coupling, $\Gamma_S \approx \Gamma_D$) and 2 (for strongly asymmetric coupling, $\Gamma_S \gg \Gamma_D$ or *vice versa*), and will alternate along with Ω for adjacent charge states. The current rectification observed in our experiments cannot be explained if the electron-electron interactions are ignored (as within the non-interacting Landauer approach) or in the case of strong coupling between the molecule and the electrodes (where the energy uncertainty associated with the lifetime of the electronic states is greater than the energy required to change the charge state of the molecule).

Comment 12:

Pages 8-12 and Fig. S22

The authors compare with Marcus-Jortner and low-temperature corrected Marcus theory in the SI, where the better performance of their method compared to the latter is declared but not clear at all from Fig. S22. Also, declaring the better performance of a method over another for graphs where, for example, I is in nA and ΔI is in pA requires much more caution at least, all the more considering that the authors are using continuous models to describe sets of discrete experimental points collected during voltage ramps.

Reply 12: Note Fig. S22 is now Fig. S23 in the new version of the manuscript. We refer to it as Fig. S23 in our reply.

The Reviewer is correct in pointing out that for some of the devices shown in Fig. S23 the superiority of our theoretical model is not very clear. In particular, the low-temperature-corrected Marcus theory appears to perform as well as our more complex approach. We note however that, the low-temperature-corrected Marcus can significantly overestimate electric current off resonance which is not shown in Figure S23. (This stems from the fact the low-temperature correction present within this approach incorrectly induces a symmetric broadening of the electron transfer rates around $\epsilon = \epsilon_0 \pm \lambda$. In reality, nuclear tunnelling effects should be much more pronounced in the Marcus inverted region.) The comparison of the full stability diagrams in Figure S24 shows this effect. It is the combination of the results in both Figures S23 and S24 taken together that emphasise the benefits of the quantum model. We believe our discussion of the results shown in Figures S23 and S24 was too brief and over-simplified, and therefore we have rewritten this section in the SI (removing terms such as ‘significantly outperforms’) with the aim of clarifying the above points.

From (SI):

Fig. S22 shows the experimental IV traces on resonance as well as the theoretical fits to the QME approach (from the main body of this work), LTC-MT and Marcus-Jortner theory. LTC-

MT yields almost as good fits as the (more sophisticated) QME approach. On the other hand, the Marcus-Jortner approach tends to perform worse than the above methods and may give rise to artefacts akin to those of the usual Marcus treatment (it still performs better than the conventional Marcus approach although at the cost of two additional fitting parameters). The full stability diagrams (measured experimentally and calculated from the parameter extracted through the above fits) are shown in Fig. S23. In contrast to the full QME approach, the LTC-MT overestimates the degree of vibrationally-induced broadening of the IV characteristics off resonance, in agreement with earlier predictions.⁶ The performance of Marcus-Jortner approach is comparable to that of the conventional Marcus theory as it again predicts early plateaus in the IV characteristics, c.f. Fig. S20. Overall, the full QME treatment significantly outperforms the remaining approaches, although it should be noted that LTC-MT and Marcus-Jortner theory both partially rectify the artefacts of the conventional Marcus approach.

To:

Fig. S23 shows the experimental IV traces on resonance as well as the theoretical fits to the quantum approach (from the main body of this work), LTC-MT and Marcus-Jortner theory. LTC-MT yields comparable fits to the (more sophisticated) quantum approach. On the other hand, the Marcus-Jortner approach tends to, in general, perform worse than the above methods as it gives rise to artefacts akin to those of the usual Marcus treatment. The artefacts are visible most clearly in 5 out of the 12 devices – D, F, G, I, and L. Despite this, it still performs better than the conventional Marcus approach, at the cost of two additional fitting parameters.

We then use the parameters from these fits to reconstruct the full stability diagrams and compare these to experimental data, this is shown in Fig. S24. Despite reproducing the IV traces well, LTC-MT overestimates the degree of vibrationally-induced broadening of the IV characteristics off resonance, in agreement with earlier predictions.⁶ This can be seen most evidently in the stability diagrams of devices such as B, C, F, G and H. The performance of Marcus-Jortner approach is comparable to that of the conventional Marcus theory as it again predicts early plateaus in the IV characteristics, c.f. Fig. S21.

Overall, Marcus-Jortner and LTC-MT approaches both have drawbacks when fitting certain devices to IVs traces and stability diagrams respectively, as described above. Therefore, by considering both Figures S23 and S24, we conclude our quantum approach provides the most robust description of the device B–M dataset.

We would also direct the Reviewer to the following wording change regarding the additional devices in the Discussion (Results, High-bias, page 12).

From:

As shown in the SI, such approaches partially rectify some of the shortcomings of MT, highlighting the non-classical mechanism of electron transfer even in these relatively high-temperature conditions. Our experimental data, however, are best described by our fully quantum mechanical treatment involving both inner and outer sphere reorganisation, as discussed above.

To:

As shown in the SI, for some devices such approaches rectify the limitations of MT, (at the expense of additional fitting parameters) and highlight the non-classical mechanism of electron transfer in these relatively high-temperature (for single-molecule devices) conditions. Our experimental data, however, are generally better described by our fully quantum mechanical treatment involving both inner and outer sphere reorganisation, as discussed above.

Other changes

The whole manuscript has been thoroughly re-checked and some sentences have been reworded to improve clarity.

REVIEWERS' COMMENTS:

Reviewer #2 (Remarks to the Author):

I feel that the authors have dealt with the comments in a satisfactory way; I recommend publication of the paper.

Reviewer #5 (Remarks to the Author):

The manuscript was considerably improved compared to the previous version, and therefore I see it much closer to be publishable in Nature Communications. However, I strongly suggest that the Authors consider the two remaining comments below prior to publication.

Eqs 2-3, pertinent discussion and results.

The Authors should further reflect on the argument in support of the omega-containing prefactors, hence on such prefactors, considering (a) the fermionic nature of the (indistinguishable) electrons, which reflects on their overall wave function, (b) the absence of a privileged spin direction in the system, and (c) the fact that the spin state of the electron on the molecule is not measured and therefore not determined. If one does so (e.g., consider the $N/N+1$ case) and takes into account that two spin states are free to receive the electron in the other electrode, I doubt that the authors can obtain different factors for the two electron transfer steps in a given electrode-molecule-electrode. The electron can be provided in two ways, but also delivered in two ways. Therefore, the argument of the Authors does not convince me.

In addition to the above, please let me stress that the rectification considered in many of the previous studies does not stem from an asymmetric potential distribution in the junction, but from different 'left' and 'right' coupling strengths.

Figure 4 and pertinent discussion

The differences between the Marcus model (orange) and quantum model (green) results compared to the experimental data in Figure 4c are, overall secondary ones. I wonder how these differences can be relevant to the description of a real device, all the more considering that the molecular bridge anyway fluctuates somewhat and the I-V trace changes from one realization to another. I agree that the formalism of the authors is more accurate than using Marcus equations, over a wider range of temperatures, and this is even more evident in Figure 4e (it is also evident in Figure 3b, but the absolute values would all be small). However, I really expect appreciable-to-significant differences at lower temperatures than 77 K. The authors should frankly state that the differences in Fig. 4c are not significant in terms of interpreting the response of a device, although differences can be more significant at lower temperatures. Therefore, the "inadequacy" of Marcus model should be restated in terms of lower accuracy in most parts of this paper.

I consider the work of the authors as a good and useful step towards detailed understanding of molecular junctions, although the particular advantages of their refined picture to interpret most experiments of practical relevance does not emerge from this study. Yet, future studies (for example, in quantum computing) could reveal the currently not evident/significant advantages of the authors' approach.

Reply to Reviewers' Comments and Details of Final Revisions

Reviewer #2 (Remarks to the Author):

Comment:

I feel that the authors have dealt with the comments in a satisfactory way; I recommend publication of the paper.

Reply: *We thank the Reviewer for taking the time to review the manuscript, and we are grateful for their recommendation of publication.*

Reviewer #5 (Remarks to the Author):

The manuscript was considerably improved compared to the previous version, and therefore I see it much closer to be publishable in Nature Communications. However, I strongly suggest that the Authors consider the two remaining comments below prior to publication.

Comment 1:

The Authors should further reflect on the argument in support of the omega-containing prefactors, hence on such prefactors, considering (a) the fermionic nature of the (indistinguishable) electrons, which reflects on their overall wave function, (b) the absence of a privileged spin direction in the system, and (c) the fact that the spin state of the electron on the molecule is not measured and therefore not determined. If one does so (e.g., consider the $N/N+1$ case) and takes into account that two spin states are free to receive the electron in the other electrode, I doubt that the authors can obtain different factors for the two electron transfer steps in a given electrode-molecule-electrode. The electron can be provided in two ways, but also delivered in two ways. Therefore, the argument of the Authors does not convince me.

In addition to the above, please let me stress that the rectification considered in many of the previous studies does not stem from an asymmetric potential distribution in the junction, but from different 'left' and 'right' coupling strengths.

Reply 1: *We thank the Reviewer for this point, and are happy to clarify. Let us also consider the $N/N+1$ case where the N -state corresponds to the non-degenerate neutral ground-state (with the overall spin of zero). Then, there exist two degenerate many-body states corresponding to the $N+1$ charge state (with the overall spin of $+1/2$ and $-1/2$). As the Reviewer correctly points out, the spin state of the electron on the molecule is not measured. Regardless of that, however, the $N+1$ charge state is two-fold (spin) degenerate.*

Since we are interested in the regime of weak molecule-lead coupling, we can consider the rate equations for the populations of these two states, $P\uparrow$ and $P\downarrow$. Due to the absence of a privileged spin direction (as pointed out by the Reviewer): $P\uparrow = P\downarrow$. This set of rate equations is now solved in a new section that we have added in the Supplementary Note 2, and, as demonstrated therein, leads to an expression for current used in the main body of this work.

The asymmetry that arises from the expression for current is due to both the omega-containing prefactors and the different left and right coupling strengths. The expression we use predicts that the

current rectification should vary between 1 ($\Gamma_S = \Gamma_D$) and 2 ($\Gamma_S \gg \Gamma_D$, or vice versa) and should alternate between adjacent charge states. This is borne out by the experimental data, where we observe rectification ratios of 1–2 for all devices, and for those devices which display 2 charge transitions (see device **A**, in Supplementary Figure 15) and device **L** (Supplementary Figure 23 and 24) the rectification ratio flips between consecutive transitions. This behaviour is captured by our model, and requires the pre-factors. Therefore we are confident in our description of the current. The Supporting Publication that was uploaded with the previous version has now been published (Nanoscale, **2019**, 11, 14820-14827), and deals with the issue of the pre-factors in much more depth.

We direct the reader to the new section in the Supplementary Information and the relevant publication by modifying the following text (Introduction, page 3):

From:

The number of possible transitions is accounted for by setting Ω to 0 for the $N/N+1$ transition or 1 for the $N-1/N$ transition.

To:

When only a single spin-degenerate level is involved in transport then the number of possible transitions is accounted for by setting Ω to 0 for the $N/N+1$ transition or 1 for the $N-1/N$ transition, as discussed in Supplementary Note 2 and in detail elsewhere⁹.

Comment 2:

The differences between the Marcus model (orange) and quantum model (green) results compared to the experimental data in Figure 4c are, overall secondary ones. I wonder how these differences can be relevant to the description of a real device, all the more considering that the molecular bridge anyway fluctuates somewhat and the I-V trace changes from one realization to another. I agree that the formalism of the authors is more accurate than using Marcus equations, over a wider range of temperatures, and this is even more evident in Figure 4e (it is also evident in Figure 3b, but the absolute values would all be small). However, I really expect appreciable-to-significant differences at lower temperatures than 77 K. The authors should frankly state that the differences in Fig. 4c are not significant in terms of interpreting the response of a device, although differences can be more significant at lower temperatures. Therefore, the “inadequacy” of Marcus model should be restated in terms of lower accuracy in most parts of this paper.

I consider the work of the authors as a good and useful step towards detailed understanding of molecular junctions, although the particular advantages of their refined picture to interpret most experiments of practical relevance does not emerge from this study. Yet, future studies (for example, in quantum computing) could reveal the currently not evident/significant advantages of the authors' approach

Reply 2: As the title suggests our study is focussed on developing an understanding of electron transfer on the single-molecule level. Therefore, although the Reviewer is correct that the data in Figure 4c (now Figure 4a) show errors that are not always large compared to the absolute magnitude of current, the data can in principle be fitted to other mathematical models, including some without physical underpinning, with small errors. The advantage of our approach is that, unlike the MT model for which the parameters are unphysical at 77 K (as outlined in the arguments presented in Results – High Bias [pages 13 & 14]), we are able to avoid this by using the quantum model.

Our results show that porphyrins contain a broad spectrum of vibrational modes (shown in Figure 4b) that are coupled to electron transfer, however one could envisage highly symmetric structures (e.g.

linear polyynes) in which only one or two vibrational modes are coupled. The molecular density of states for these structures would be far less well represented by a Gaussian lineshape than that of the porphyrin molecules studied here, and therefore the temperature at which MT and quantum mechanical descriptions converge in their accuracy could extend to even higher temperatures than 77 K. This would lead to even less physical values of λ and the coupling to the electrodes. In contrast our model is robust to changes in molecular structure, and therefore, as the Reviewer points out, future studies could reveal situations in which the advantages of our approach are more significant than presented here.

Overall we concede the term 'inadequacy' is not necessarily the best term to use, as it depends on the purpose of the device. Therefore as per the requests of the Reviewer it has been removed. We have changed the following text (Discussion, page 15):

From

We have further shown that at cryogenic temperatures (below 77 K), Marcus theory also constitutes an inadequate description of the charge transport mechanism due to the importance of nuclear tunnelling under these conditions.

To:

We have further shown that at cryogenic temperatures (below 77 K), Marcus theory constitutes a less accurate description of the charge transport mechanism due to the importance of nuclear tunnelling under these conditions.

and altered the following sentences (Results, page 12) to comply with the Reviewers request for a frank statement regarding Figure 4c (now Figure 5a)

From:

We find that our experimental charge transport data for devices **B–D** can be described by MT with appreciable errors in the fits compared to the magnitude of the current, as shown in Figure 4c²³, and there are features of the charge transport that are not captured by this approach.

To:

We find that the experimental charge transport data for devices **B–D** at 77 K can be described by MT since at this temperature the errors in the fits of the *IV* characteristics are not particularly large compared to the magnitude of the current (as shown in Figure 5a). However, there are features in the data that are not captured by this approach that we must explain if we wish to develop a detailed and physical understanding of the mechanism of charge transport that is valid over a wide temperature range and robust to changes in the molecular structure.

Overall, we believe we have addressed the Reviewers concerns fully, and hope they now support publication.